# Genome-wide analysis of cardiac ventricular phenotypes reveals novel loci and therapeutic targets for heart failure

Hannah L. Nicholls [1], Jose D. Vargas[1,2,3], Mihir M. Sanghvi[1,4], Hyo-Suk Ahn [5], C. Anwar A. Chahal [1,4,6,7], Mohammed Y. Khanji[1,4], Steffen E. Petersen [1,4], Patricia B. Munroe [1] & Nay Aung [1,4] ✉

Left and right ventricular imaging measures are essential for heart failure diagnosis and prognostication, yet their genetic architecture remains under-explored. We conduct genome-wide association analyses of twenty left and right cardiovascular magnetic resonance phenotypes in 56,509 UK Biobank participants, including conventional measurements (e.g., volumes/ejection fraction) and novel parameters (left ventricular global function index and myocardial contraction fraction). We identify 200 loci associated with at least one phenotype ($P < 5 \times 10^{-8}$); 58 being novel. A polygenic risk score for left ventricular global function index negative associates with heart failure in phenome-wide scan. Rare variant analysis reveals enrichment of deleterious variants across 13 genes ($P < 2.5 \times 10^{-6}$). Colocalisation with heart failure implicates 23 shared loci and bioinformatic analysis prioritises genes including *HSPB7, CAMK2D, ALDH2, ENG*, and *YWHAE*. Druggability analysis highlights *PDE3A*, informing divergent effects of non-selective PDE3 inhibition. In this work, we expand our knowledge of cardiac ventricular genetics, suggesting potential heart failure therapeutic targets.

Aberrations in cardiac ventricular structure and function underlie heart failure (HF), a leading cause of mortality and morbidity worldwide[1]. Cardiac imaging phenotypes capture complementary aspects of cardiac anatomy and physiology, including chamber size, myocardial hypertrophy and contractile performance, and are key measures used in early diagnosis of cardiac abnormalities. Left ventricular (LV) volumes represent the global size of the main pumping chamber, with end-diastolic volume reflecting preload and chamber dilatation, and end-systolic volume reflecting residual blood after contraction and systolic efficiency. LV mass quantifies myocardial hypertrophy and remodelling, while LV stroke volume and LV ejection fraction (LVEF) represent pump function and systolic performance. Notably, LV volume, LV mass, and LV systolic function, measured by LV

ejection fraction, are instrumental in HF diagnosis and prognostication. Recently, novel LV functional measures, LV global function index (LVGFI) and LV myocardial contraction fraction (LVMCF), have demonstrated superior predictive capabilities for adverse cardiovascular outcomes by integrating anatomic information in estimating global systolic performance[2,3]. In parallel, right ventricular (RV) structure and function have independent and incremental roles in determining HF[4-6], with RV volumes, stroke volume, and ejection fraction reflecting pulmonary vascular loading, RV contractile function, and ventricular interdependence.

Growing evidence suggests these biomarkers, and their genetic underpinnings, may uncover mechanisms driving cardiovascular outcomes. Recent genome-wide association studies (GWAS) of

[1]William Harvey Research Institute, Queen Mary University of London, London, UK. [2]Veterans Affairs Medical Center, Washington, DC, USA. [3]Georgetown University, Washington, DC, USA. [4]Barts Heart Centre, St Bartholomew's Hospital, Barts Health NHS Trust, West Smithfield, London, UK. [5]The Catholic University of Korea, Uijeongbu St. Mary's Hospital, Seoul, Korea. [6]Center for Inherited Cardiovascular Diseases, WellSpan Health, York, PA, USA. [7]Department of Cardiovascular Diseases, Mayo Clinic, Rochester, MN, USA. ✉e-mail: n.aung@qmul.ac.uk

cardiovascular magnetic resonance (CMR)-derived ventricular endo-phenotypes reported susceptibility loci implicating genes for myocardial contractility, cellular adhesion and Mendelian cardiomyopathy genes. However, these loci explained a fraction of the estimated trait heritability. These studies focused on established CMR phenotypes (e.g., LV/RV volumes with recent study sample sizes ranging from approximately 5000[7] to 43,000[8]), highlighting the need for larger studies including emerging CMR traits. These studies also have not utilised dense imputation panels like Trans-Omics for Precision Medicine (TOPMed), nor have they fully explored rare variants' contributions. CMR, the gold standard for assessing cardiac size and function, benefits from a high signal-to-noise ratio and avoidance of geometric assumptions. The UK Biobank's CMR imaging, alongside densely imputed genotype and whole-exome sequencing (WES), enables robust investigations of the genetics of ventricular phenotypes. This effort is supported by artificial intelligence segmentation algorithms proven to be the most precise and reproducible method to derive imaging measurements[9]. Moreover, resources like Open Targets enable in-depth druggability analysis between CMR loci and HF drug targets. Recent HF therapeutic research diverges from the conventional neurohormonal axis to newer pathways such as renal glucose excretion (sodium glucose cotransporter 2 inhibitors), nitric oxide synthase-guanylate cyclase system (Vericiguat), and myocardial contractility (Omecamtiv Mecarbil)[10]. Understanding the shared genetic basis of ventricular endophenotypes and HF could reveal HF drug target pathways and drug repurposing opportunities.

Here, we systematically evaluate the genetic architecture of clinically relevant LV and RV imaging traits, including novel LV functional parameters, by leveraging TOPMed imputed genotypes and WES in up to 56,509 European UK Biobank individuals. GWAS performed on 20 LV and RV traits discovered 200 loci, 58 being novel CMR loci. 23 loci colocalised and shared causality with HF loci. Downstream bioinformatic analysis prioritised several candidate genes underpinning the cardiac remodelling process (e.g., *ADAMTSL3, STRN, HSPB7, CAMK2D, IGF1R, ALDH2, ENG*, and *YWHAE*). Druggability analysis identified 59 prioritised genes with gene-drug interactions. Rare variant analysis revealed 13 significantly associated genes, which include cardiomyopathy genes (e.g. *TTN, RBM20*) and genes with indicated functional roles in cardiomyopathies (*CORO6, DUSP13, GAA, GIP, CYP3A4*, and *SIRT4*). Phenome-wide association study (PheWAS) of LV and RV polygenic risk scores (PRS) revealed significant associations with various cardiovascular conditions. Novel phenotypes, LVGFI and LVMCF, showed negative associations with cardiovascular diseases and a positive association with coeliac disease. These findings significantly advance our knowledge of LV and RV genetic architecture and provide evidence for potential HF drug targets.

## Results

### Study cohort
The UK Biobank participants' characteristics are detailed in Supplementary Data 1. In total, 56,509 European individuals with CMR studies (mean age of 64 years, 48% men) were included. LV and RV measurements were derived by deep-learning-based segmentation. These included LV end-diastolic volume (LVEDV), LVEDV indexed to body surface area (BSA) (LVEDV BSA), LV end-systolic volume (LVESV), LVESV BSA, LV stroke volume (LVSV), LVSV BSA, LV mass (LVM), LVM BSA, LV mass to end-diastolic volume ratio (LVMVR), LV ejection fraction (LVEF), LV global function index (LVGFI), LV myocardial contraction function (LVMCF), RV end-diastolic volume (RVEDV), RVEDV BSA, RV end-systolic volume (RVESV), RVESV BSA, RV stroke volume (RVSV), RVSV BSA, RV ejection fraction (RVEF), and RVEDV to LVEDV ratio (RV-LV ratio). The sample size for each trait ranged from 56,204 to 56,231. The study design is summarised in Fig. 1.

### Trait heritability and correlation structure
Single-nucleotide polymorphism (SNP)-based heritability of LV and RV traits ranged from 18.3% (LVGFI) to 35.8% (LVMVR) (Supplementary Fig. 1). The novel CMR phenotypes, LVGFI and LVMCF, had a heritability of 18.3% and 24.7%, respectively.

### Genome-wide discovery analysis
We conducted GWAS for 8.9 million TOPMed-imputed variants with minor allele frequency (MAF) ≥ 0.01 and imputation INFO >0.3. We first performed single-trait GWAS for each phenotype. Then, to increase power, we conducted pairwise multi-trait discovery analyses by combining genetically correlated phenotypes ($r_g$ > 0.7), allowing detection of additional loci associated with shared genetic variation. In total, 200 loci were identified across GWAS of 20 LV and RV phenotypes (Supplementary Data 2). These comprised 45 loci for LVEDV, 64 for LVEDV BSA, 49 for LVESV, 65 for LVESV BSA, 31 for LVSV, 26 for LVSV BSA, 41 for LVM, 46 for LVM BSA, 67 for LVMVR, 44 for LVEF, 26 for LVGFI, 41 for LVMCF, 35 for RVEDV, 39 for RVEDV BSA, 38 for RVESV, 38 for RVESV BSA, 24 for RVSV, 26 for RVSV BSA, 26 for RVEF, and 17 for RV-LV ratio at conventional GWAS $P < 5 \times 10^{-8}$. In total, 177 loci were associated with >1 phenotype (Fig. 2, Supplementary Figs. 2 and 3, Supplementary Data 3), which is anticipated given their genotypic correlations, and 23 loci showed greater specificity to individual phenotypes, signified by weaker associations with other ventricular phenotypes ($P > 1 \times 10^{-5}$). In total, 58 loci were novel compared to known CMR loci (lead variants >1 Mb distance and none being linkage disequilibrium, LD, proxies with $r^2 > 0.4$ within 4 Mb) (Supplementary Data 4). The novel phenotypes, LVGFI and LVMCF, had ten novel loci, of which two were associated with LVGFI (*LINC01249* and *RBM38*) and eight with LVMCF (*MAP9, TNFAIP8, DKFZp451BO82, NRP1, SLC16A12, LINC00477, LARP4/DIP2B* and *FURIN*). Twenty-seven loci had independent signals in conditional and joint analysis. We found limited genomic inflation (λ range: 1.01–1.138), much of which was resolved as polygenic signal (LD score regression intercept range: 1.004–1.06) (Supplementary Fig. 4)− indicating low evidence for population stratification or cryptic relatedness. The proportion of phenotypic variance explained by the lead variants combined ranged from $R^2$ of 0.52% (RVSV BSA) to 1.52% (LVESV).

### Genetic overlap with other complex traits
We assessed pleiotropic associations of lead variants and their credible sets or close proxies ($r^2 \geq 0.8$, $n = 19,116$) with traits in Phenoscanner[11] and the GWAS Catalogue[12]. In Phenoscanner, 61 ventricular loci associated with 453 traits, with LV and RV phenotypes sharing 379 traits (Supplementary Data 5 and 6). Curation of phenotypic groups identified SNPs mostly associating with blood composition, physical traits, or cardiovascular phenotypes (Fig. 3). The *ATXN2/SH2B3* locus was highly pleiotropic with significant associations with >100 traits. Variants at 48 loci showed associations with cardiovascular phenotypes. Seven loci (*SLC35F1, MYH6, TTN/PLEKHA3/FKBP7/CCDC141, NDST2/SYNPO2L, MYH7, NKX2-5* and *TBX5*) harboured variants associated with atrial fibrillation. Variants in nine loci (*VEGFA, BRAP/ATXN2/SH2B3/HECTD4/PTPN11/MAPKAPK5-AS1, GOSR2, RSPH6A, NOS3, PDE5A, KIAA1462/JCAD, MAP3K7CL, CENPW/MIR588*) were associated with coronary artery disease. Variants in three loci were associated with dilated cardiomyopathy (*MAP3K7CL, C1orf64/CLCNKA/ZBTB17/HSPB7* and *BAG3*).

In comparison, 100 associations were found in the GWAS catalogue with 326 traits, 49 being cardiovascular phenotypes (Supplementary Data 7). Highlighting those not in Phenoscanner, four loci (*TIAL1, ATXN2, TBX3* and *BAG3*) were associated with all-cause HF and four loci were associated with hypertrophic cardiomyopathy (HCM) (*VTI1A, BAG3, ADPRHL1* and *TBX3*).

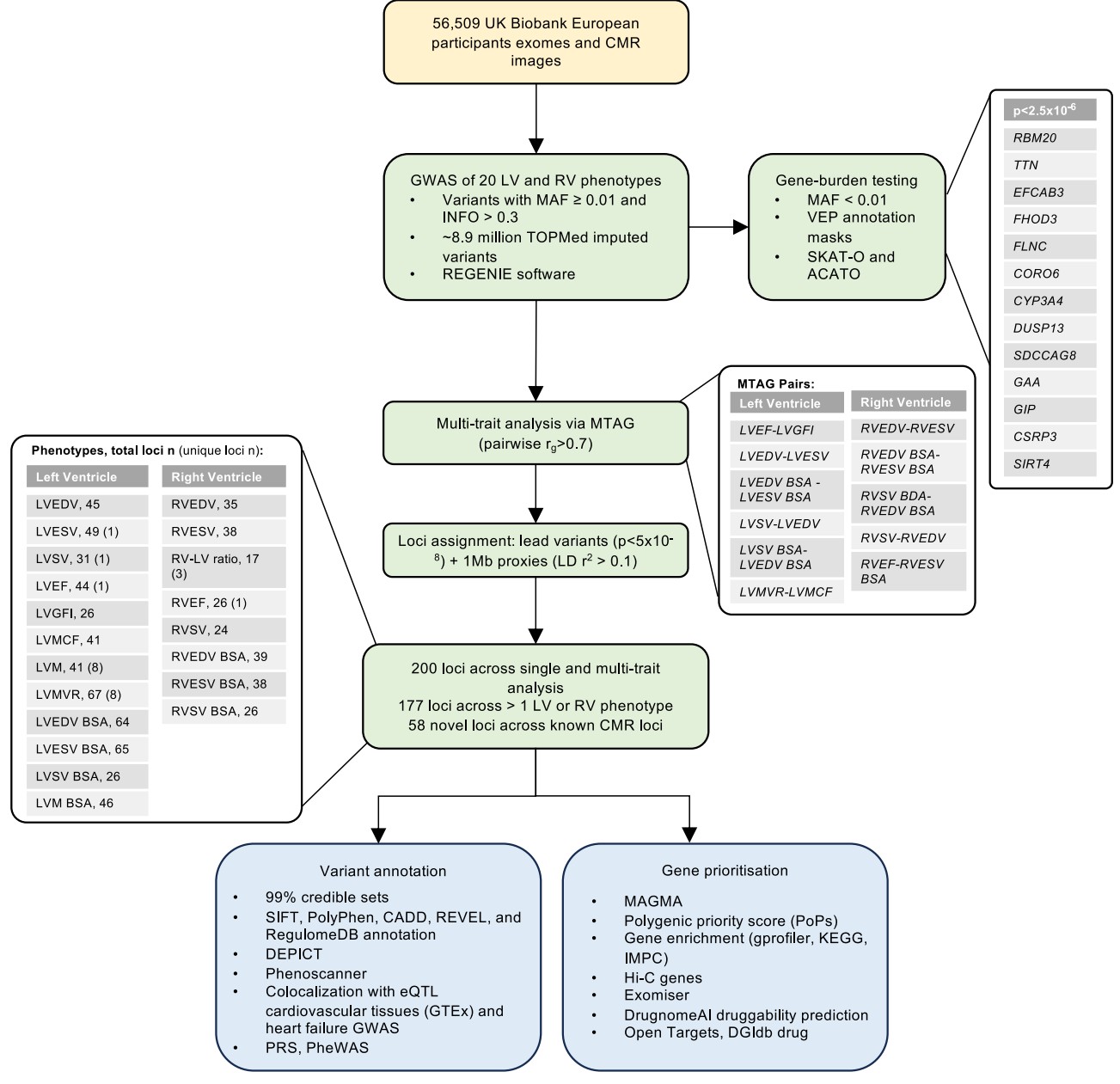

**Fig. 1 | Flowchart of analysis for left and right ventricular genome-wide association study.** GWAS genome-wide association study, CMR cardiovascular magnetic resonance, LV left ventricle, RV right ventricle, RVEDV right ventricular end-diastolic volume, RVESV right ventricular end-systolic volume, RVSV right ventricular stroke volume, RVEF right ventricular ejection fraction, EA effect allele, NEA non-effect allele, SE standard error, RVEDV right ventricular end-diastolic volume, RVESV right ventricular end-systolic volume, RVSV right ventricular stroke volume, RVEF right ventricular ejection fraction, RV LV ratio right ventricular to left ventricular size ratio; RVEDV BSA right ventricular end-diastolic volume indexed to body surface area, RVESV BSA right ventricular end-systolic volume indexed to body surface area, RVSV BSA right ventricular stroke volume indexed to body surface area, LVEDV left ventricular end-diastolic volume, LVESV left ventricular end-systolic volume, LVEF left ventricular ejection fraction, LVSV left ventricular stroke volume, LVM left ventricular mass, LVMVR left ventricular mass to volume ratio, LVGFI left ventricular global function index, LVMCF left ventricular myocardial contraction fraction, LVEDV BSA left ventricular end-diastolic volume indexed to body surface area, LVESV BSA left ventricular end-systolic volume indexed to body surface area, LVSV BSA left ventricular stroke volume indexed to body surface area; LVM BSA left ventricular mass indexed to body surface area, MAF minor allele frequency, INFO imputation quality score, LD linkage disequilibrium, MTAG multitrait analysis of genome-wide association, SIFT sorting intolerant from tolerant, PolyPhen-2 polymorphism phenotyping 2, CADD combined annotation-dependent depletion, DEPICT data-driven expression prioritised integration for complex traits, eQTL expression quantitative trait loci, GTEx genotype-tissue expression, PRS polygenic risk scores, MAGMA multimarker analysis of genomic annotation, KEGG Kyoto Encyclopaedia of Genes and Genomes, IMPC International Mouse Phenotyping Consortium, Hi−C long-range chromatic interaction, DGIdb drug–gene interaction database.

## Characteristics of GWAS variants

Among 19,116 variants in credible sets or close proxies to lead variants ($r^2 \geq 0.8$), 0.65% were exonic variants, 45.35% intronic variants, 5.88% intergenic variants, and the remaining were non-coding RNA, untranslated and upstream/downstream variants (Supplementary Data 8). Among exonic variants, 27 variants across 19 loci (annotated to 21 genes) were found to have damaging consequences (Methods). Amongst the genes, four are known cardiomyopathy genes (e.g. *TTN, MYO18B, FLNC* and *BAG3*), seven have indicated roles in HF/cardiac remodelling (*CCDC141*[13], *MAPT*[14], *MAP4*[15], *SYNPO2L*[16], *ADPRHL1*[17], *TNXB*[18]

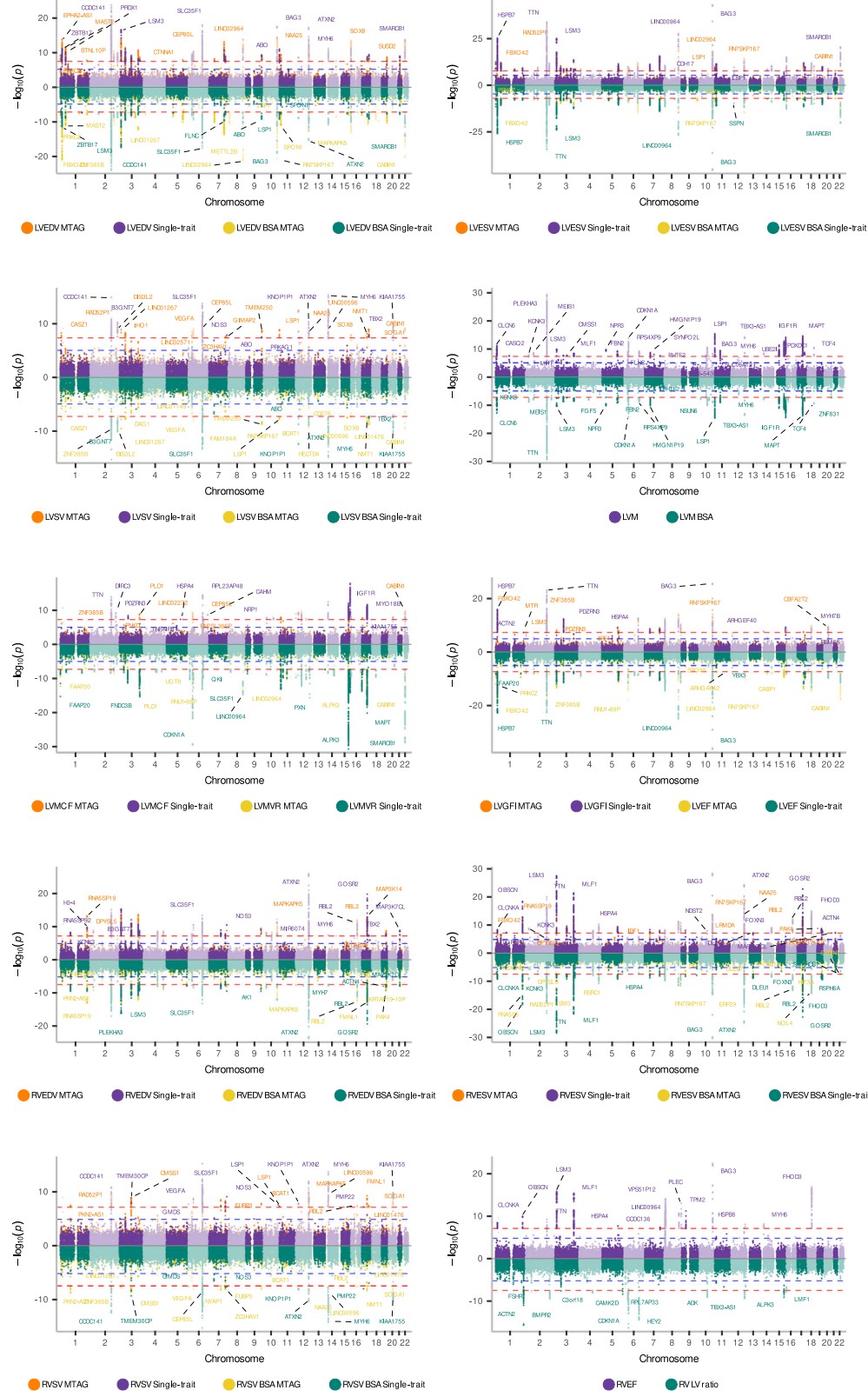

and *OBSCN*[19]), and ten have potential cardiovascular impacts (*MAPT-AS1, PROB1, NOL9, HCG22, SRL, FERMT2, ARHGEF40, KIAA1755, SPATS2L* and *B3GNT7*−with *FERMT2* in a novel CMR locus). Colocalisation experiments using GTEx (Genotype-Tissue Expression) data in cardiovascular tissues suggested 14 causal variants in eight loci with eight nearby genes (*PDZRN3, PROM1, SPON1, LINC00964, PLEC, ACTN2, HEY2* and *ERBB4*−all known CMR loci). All genes, except *ACTN2, ERBB4* and

*PLEC*, colocalised with >1 cardiovascular tissue−with *ACTN2, ERBB4* and *PLEC* colocalising with atrial appendage, aorta and LV, respectively. Two of the eight loci identified unique causal variants per phenotype: LVESV (rs1969539, *SPON1*, colocalising with LV and atrial appendage) and RV LV ratio (rs12724121, *ACTN2*, colocalising with atrial appendage). Six loci had a causal variant different from the lead variant. Data-driven expression prioritised integration for complex traits

**Fig. 2 | Miami plots for all left and right ventricular traits.** Each point represents a single-variant association tested using a two-sided Wald test from a linear regression under an additive genetic model. Points are colour-coded to identify phenotype traits across single-trait (purple and green points) and multi-trait (orange and yellow points) analysis. The red line indicates the genome-wide significant threshold at $P < 5 \times 10^{-8}$, accounting for multiple testing across the genome. The blue line indicates the suggestive genome-wide threshold of $P < 1 \times 10^{-5}$. The y-axis denotes the $\log_{10}$(p-value), and the x-axis denotes the chromosomes. Source data are provided as a Source Data file. LVEDV left ventricular end-diastolic volume, LVESV left ventricular end-systolic volume, LVSV left ventricular stroke volume, LVEF left ventricular ejection fraction, LVGFI left ventricular global function index, LVMCF

left ventricular myocardial contraction fraction, LVM left ventricular mass, LVMVR left ventricular mass-to-volume ratio, LVEDV BSA left ventricular end-diastolic volume indexed to body surface area; LVESV BSA left ventricular end-systolic volume indexed to body surface area, LVSV BSA left ventricular stroke volume indexed to body surface area, LVM BSA left ventricular mass indexed to body surface area, RVEDV right ventricular end-diastolic volume, RVESV right ventricular end-systolic volume, RVSV right ventricular stroke volume, RVEF right ventricular ejection fraction, RV LV ratio right ventricular to left ventricular ratio, RVEDV BSA right ventricular end-diastolic volume indexed to body surface area, RVESV BSA right ventricular end-systolic volume indexed to body surface area, RVSV BSA right ventricular stroke volume indexed to body surface area.

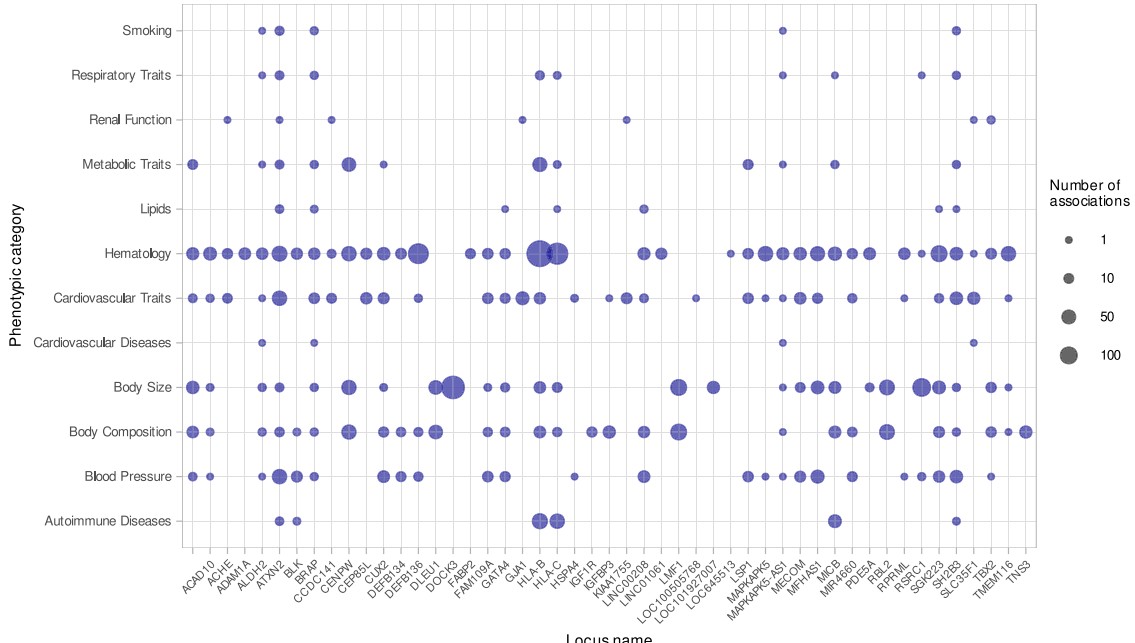

**Fig. 3 | Pleiotropic associations between left and right ventricular variants in Phenoscanner.** The top 50 loci with the highest counts of pleiotropic associations across manually curated phenotype groupings. The y-axis shows the phenotype groups, and the x-axis denotes the loci by locus gene names. The number of associated variants per phenotype group and locus is denoted by plot point size. Source data are provided as a Source Data file.

(DEPICT) identified 1440 associations with the most significant ones being thick ventricular wall ($P = 2.58 \times 10^{-11}$), cellular substrate adherens junction ($P = 3.04 \times 10^{-11}$), and substrate junction ($P = 8.13 \times 10^{-11}$) (Supplementary Data 9). Significant descriptions included FHL2 and HIF-1α protein interactions with P-values of $1.19 \times 10^{-10}$ and $2.19 \times 10^{-9}$, respectively, emphasising roles in mechanical stress and hypoxia. FHL2 influences cellular motility and structural integrity, impacting the progression of HCM and potentially dilated cardiomyopathy[20]. Meanwhile, HIF-1α regulates response to hypoxia and was shown to be associated with glucose-induced cardiomyocyte hypertrophy[21].

### Relationship with HF genetics
We investigated the shared genetic architecture of ventricular traits and all-cause HF by pairwise colocalisation with a HF GWAS from Levin et al.[22]. (Supplementary Data 10). Twenty-three ventricular loci colocalised with HF loci (locus level H4 > 0.8), with two being novel CMR loci (*CASZ1* and *SLC4A7*). Among these loci, 9 were known HF loci (*FLJ37453/HSPB7/C1orf64/ZBTB17/CLCNKA, ABO, BAG3/TIAL1, ATXN2/ BRAP/ MAPKAPK5-AS1, TBX3, CDKN1A, CDKN2C, STRN, CDKN1A*) with HF GWAS P-value $< 5 \times 10^{-8}$. The remaining 14 loci were not within 1 Mb of HF lead SNPs, with each achieving HF P-values ranging from $2.4 \times 10^{-4}$ to $1.19 \times 10^{-7}$–including both novel CMR loci (*CASZ1* and *SLC4A7*) and known CMR loci (*MIR4662B/LINC00964, LSP1, YWHAE, RPRML/GOSR2, GSTTP2/SMARCB1/DERL3, CAND2, VTI1A, ALPK3,*

*ACTN2, KCNK3, CAMK2D, MAPRE2,*), highlighting the value of imaging traits to uncover potential genetic determinants of HF through colocalisation. Nine loci colocalised to different causal variants (H3 > 0.8): *SPATS2L, HLA-B/HCP5/HLA-C, MYO1C, CDKN1A, NUDT3, LOC102723373/ RBL2/AKTIP, ADK, TBX3* and *NSUN6*.

### Gene prioritisation and pathway enrichment
In total, 1339 candidate genes were annotated within 10 kb of ventricular loci. These genes were annotated with polygenic priority scores (PoPS)[23] (Supplementary Data 11), with 171 genes prioritised by PoPS alone. We developed a gene prioritisation scoring system, based on the presence of evidence across multiple bioinformatic resources, grouped into four categories: (1) PoPS, (2) colocalisation with HF, (3) OMIM or ClinGen cardiovascular Mendelian genes and (4) cardiovascular mouse models, Hi-C (high-throughput chromosome conformation capture), or enriched GTEx expression. Each annotation was weighted equally, with OMIM and ClinGen considered jointly (i.e., the presence of either counts as one scored annotation), and mouse models, GTEx and Hi−C also considered jointly in the same way. Higher scores indicate stronger multi-source support for gene prioritisation (Supplementary Data 12) (Methods). For example, a score of 4 is the highest possible score (a gene has all four lines of evidence), and a score of one is the lowest (a gene only has one line of supporting evidence). In total, 488 out of 1339 genes achieved a score of at least 1.

Two genes had a score of four (*BAG3* and *ALPK3*), 28 genes scored 3, 118 scored 2, and 340 scored 1. Amongst novel CMR loci, 6 genes had scores of 3 (*SLC16A12*, *CASZ1*, *AK1*, *HMCN2*, *SLC4A7* and *ADK*). The genes that scored 3 include Mendelian cardiovascular disease genes (e.g. *FLNC*, *CASQ2*, *IGF1R* and *NOS1AP*). We investigated pathway and mouse model enrichment of all 488 prioritised genes with score ≥1 using g:Profiler and the International Mouse Phenotyping Consortium (IMPC) (Supplementary Data 13 and 14). g:Profiler identified muscle contraction (adjusted $P = 1.02 \times 10^{-13}$), striated muscle contraction pathway (adjusted $P$-value $3.76 \times 10^{-12}$), HCM (adjusted $P = 5.33 \times 10^{-07}$) and dilated cardiomyopathy (adjusted $P = 8.39 \times 10^{-06}$) as the most significantly enriched pathways (Supplementary Data 13). These genes were enriched for 190 mouse model phenotypes (93 being cardiovascular-related), with the most significant being enlarged heart (adjusted $P = 2.30 \times 10^{-55}$), abnormal heart morphology (adjusted $P = 2.58 \times 10^{-46}$), and abnormal kidney morphology (adjusted $P = 8.69 \times 10^{-36}$) (Supplementary Data 14, Supplementary Fig. 5).

### Drug repurposing or repositioning opportunities

Tractability of prioritised genes for drug development was referenced against Open Targets. 59 genes showed therapeutic potential (Supplementary Data 15). Of these genes, only six have been the focus of recent HF druggability research that is also genetically-informed by CMR phenotypes (*KCNH2*[24,25], *MAPT*[24], *TNFSF12*[18,24], *MYH7B*[24], *ATP2A2*[24] and *ALDH2*[25]). Fifteen out of 59 genes interacted with a HF indicated drug studied in clinical trials up to at least phase 2 (*ACHE* [phase 2], GUCY1A1/GUCY1A3 [phase 4], *KCNH2* [phase 4], *PDE3A* [phase 4], *IL1R1* [phase 2], *PDE5A* [phase 4], *NDUFS3* [phase 2], *ATP1B2* [phase 2], *MYH6* [phase 3], *MYH7* [phase 3], *MYH7B* [phase 3], *MYL2* [phase 3], *MYL4* [phase 3], *TNNC1* [phase 3] and *ATP2A2* [phase 2]). Twenty-one out of the 59 genes encode proteins that interact with 76 cardiovascular drugs, among which *IGF1R* was prioritised at score 3. Nine of these 21 genes interact with cardiovascular drugs exclusively (*NPR3*, *MYH6*, *MYH7*, *MYH7B*, *MYL2*, *MYL4*, *E2F8*, *TNNC1* and *NOS3*). Of these 21 genes, drug enrichment analysis using the drug repurposing hub mechanism of action database showed significant enrichment for electrophysiological and vasodilatory mechanisms (Supplementary Data 16), such as potassium-channel blockers (amiodarone, ibutilide, dofetilide, vernakalant; adjusted $P$-value $2.15 \times 10^{-7}$) and phosphodiesterase inhibitors (dipyridamole, enoximone, avanafil, milrinone; adjusted $P$-value $7.14 \times 10^{-5}$). Thirty-eight genes interact exclusively with non-cardiovascular disease drugs (three of which, *ENG*, *ALDH2* and *EPHB1*, were scored 3 by our gene prioritisation system). Twelve of the non-CVD drug-interacting genes were also significantly enriched in cardiovascular pathways (*ENG*, *ATP2A2*, *KCNK3*, *PRKCA*, *PTK2*, *PTPN11*, *NPY2R*, *NRP1*, *PRKAG1*, *ERBB4*, *PDE5A*, and *VEGFA*). *ENG*, *NRP1* and *NPY2R* were in novel CMR loci and have associations with coronary artery disease[26] and systolic blood pressure[27].

A manual review of drug interactions identified several compounds of interest, including pharmacological agents that target phosphodiesterase-III (PDE3). In our GWAS, *PDE3A* lead variant (rs10841522) was associated with higher LVMVR ($P$-value $= 8.9 \times 10^{-10}$) and a lower LVMCF ($P$-value $= 6.76 \times 10^{-6}$), indicating adverse remodelling. This variant also had significant long-range Hi−C interactions in the right ventricular tissue. While this specific variant was not found in the GTEx eQTL data, *PDE3A* was highly expressed in the GTEx cardiac LV and coronary tissues (Supplementary Data 12).

On investigating gene–drug interactions in the drug–gene interaction database (DGIdb), 55/488 genes have drug interactions not in Open Targets (Supplementary Data 12). Of these genes, four genes were scored 3 (*CAMK2D*, *CDKN1A*, *NPPA* and *NOS1AP*, with *CAMK2D* having been previously identified as a potential HF drug target[18]). We also explored genes with no drug interactions that are potentially druggable. 53 prioritised genes were annotated as druggable in DGIdb, but do not have drug targets (e.g. *MMP11*). 24/53 genes have a drugnomeAI prediction >0.5, one of which, *ACTN4* (a known CMR locus) was significantly enriched in cardiovascular mouse model phenotypes, and three were significantly enriched in cardiovascular pathways (*DAG1*, *ITGA9* and *DMPK*−with *DAG1* and *ITGA9* being novel CMR loci) (Supplementary Data 15).

### Rare variant analysis

Rare variants (MAF < 0.01) from WES data were aggregated into gene units filtered by functional masks (see Methods) for gene-burden testing. We identified 13 genes via a pre-specified p-value threshold ($P < 2.5 \times 10^{-6}$) (Supplementary Data 17). Significantly associated genes included cardiomyopathy genes (*TTN*, *RBM20*, *FHOD3*, *CSRP3* and *FLNC*). Six genes have functional research identifying cardiomyopathy roles (*CORO6*, *DUSP13*, *GAA*, *GIP*, *CYP3A4* and *SIRT4*). Two of the genes, *EFCAB3* and *SDCCAG8*, have associations and gene expression study for aortic aneurysm[28] and whole blood methylation analysis linked to cardiovascular diseases[29]. The genes have various functions, including calcium ion channel regulation (*EFCAB3*), actin filament binding (*FHOD3*, *FLNC* and *CORO6*), RNA splicing (*RBM20*), encoding a key cardiac muscle protein (*TTN*), cell proliferation and differentiation (*DUSP13* and *CSRP3*), mitosis (*SDCCAG8*), glycogen degradation (*GAA*), encoding incretin (*GIP*), drug metabolism (*CYP3A4*), cell growth (*CSRP3*) and mono-ADP-ribosyltransferase activity (*SIRT4*). We evaluated whether the rare variant associations were independent of nearby common variants by conditioning on lead variants or close proxies in the same regions. This analysis showed six rare variant gene-level associations were independent of nearby common variants in conditional analyses (*FHOD3*, *FLNC*, *RBM20*, *TTN*, *CSRP3* and *SIRT4*).

### Phenome-wide association study

PheWAS was performed using PRS calculated via PRScs[30] (Methods) and 1855 phenotypes (Supplementary Fig. 6, Supplementary Data 18). The analyses showed significant positive associations of morphological phenotypes (e.g. LVEDV, LVESV, LVM and LVMVR) with adverse cardiovascular outcomes (e.g. HF, hypertension, cardiomyopathies and cardiac dysrhythmias). Functional phenotypes (e.g. LVEF, LVMCF and LVGFI) had strong negative associations with the same cardiovascular outcomes. The novel LVGFI and LVMCF phenotypes had their strongest negative association with hypertension, with LVGFI also having strong negative associations with HF and cardiomyopathies (Fig. 4). Interestingly, the most significant non-cardiovascular association for LVGFI and LVMCF was with coeliac disease.

## Discussion

This study investigated the genetic determinants of clinically relevant ventricular phenotypes, including novel and prognostic measures, LVGFI and LVMCF, derived from highly precise and reproducible CMR imaging in the UK Biobank. Leveraging the TOPmed imputation panel, single and multi-trait analyses of -8.9 million common genetic variants from 56,509 European individuals discovered 200 susceptibility loci, of which 58 were novel CMR loci. These loci were enriched in the biological pathways for heart development and mechanical stress-related cardiac remodelling. Colocalisation with all-cause HF GWAS revealed 23 of our ventricular loci share the same causal variant as HF. Rare variant analysis found 13 significantly associated genes, 5 being inherited cardiomyopathy genes and 6 being functionally implicated in cardiomyopathy development. Bioinformatic analyses identified 28 candidate genes with strong in-silico evidence (prioritisation score ≥ 3). The PheWAS identified relationships between LV and RV phenotypes and HF, hypertension, cardiomyopathies, and cardiac dysrhythmias, emphasising the clinical relevance of these CMR phenotypes in determining cardiovascular disease prognosis.

Heritability estimates for LV and RV traits ranged from 18.3% to 35.8%, alongside moderate to high genetic correlation, reflecting

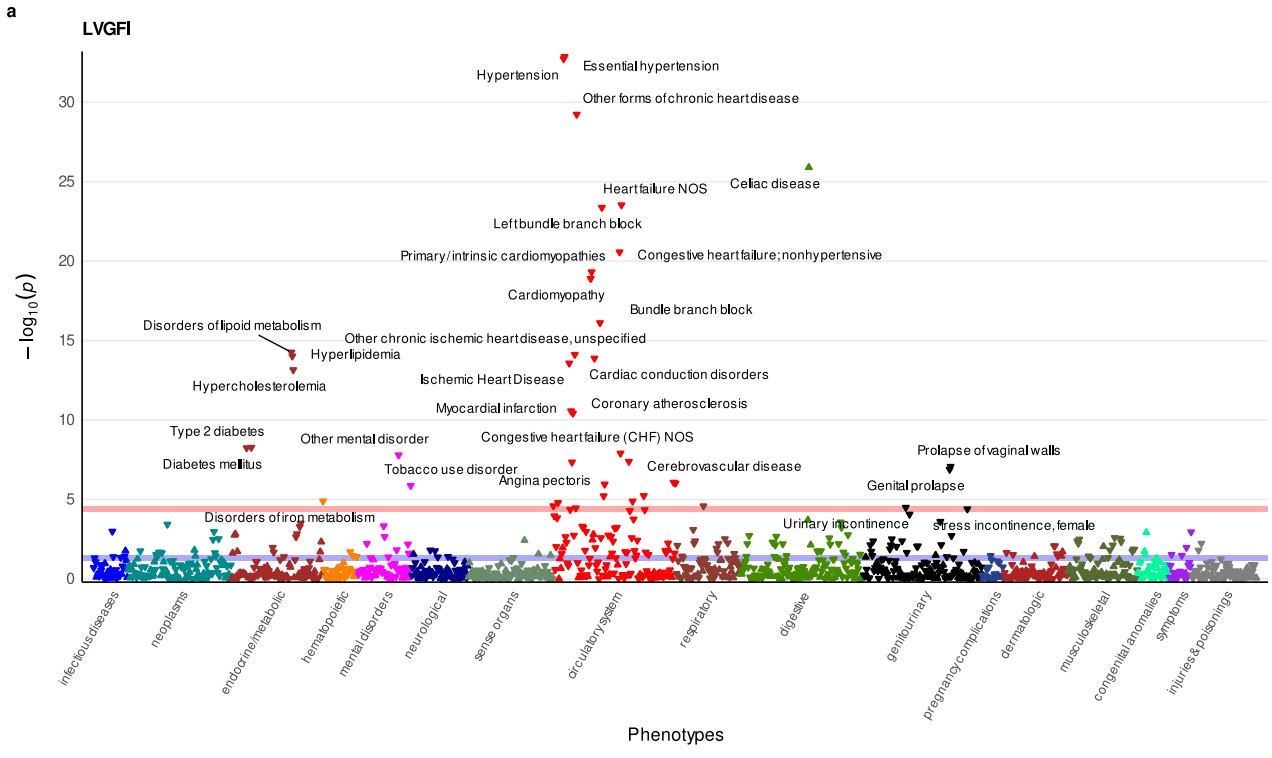

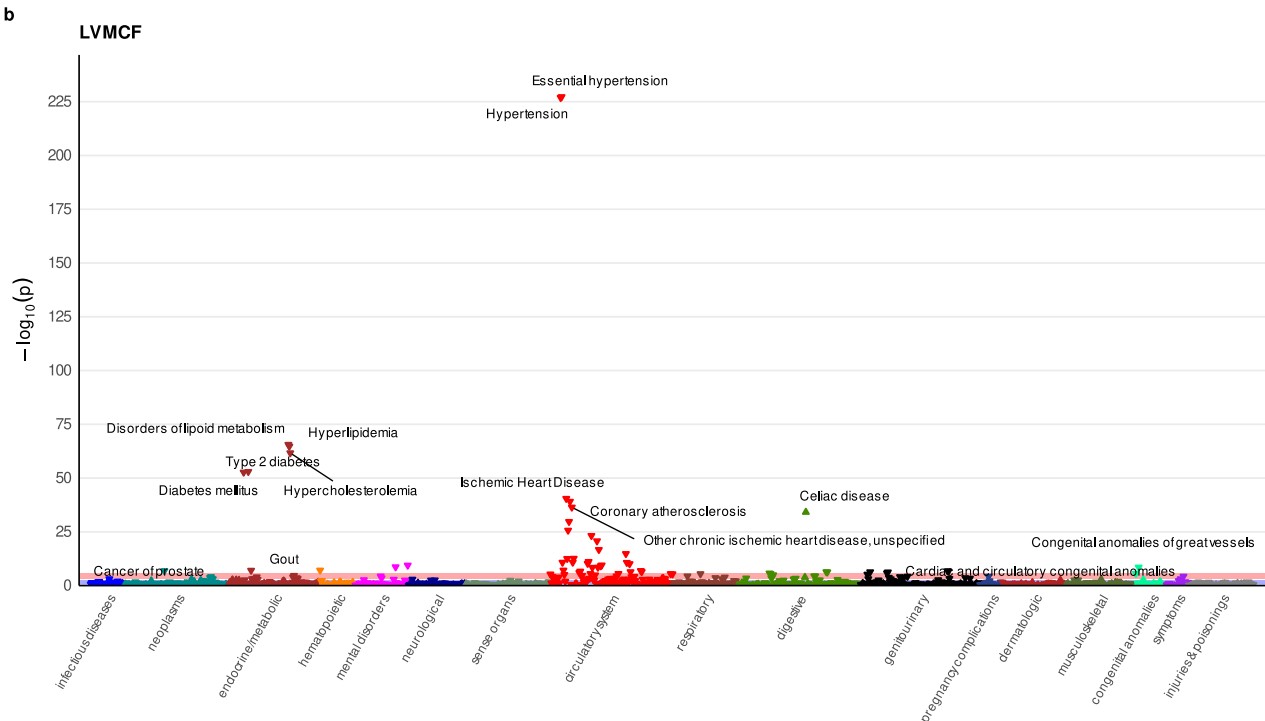

**Fig. 4 | Phenome-wide association study of novel left ventricular phenotypes.** LVGFI, left ventricular global function index (**a**), and LVMCF, left ventricular myocardial contraction fraction (**b**), were tested using two-sided regression models across phenotypes. Multiple testing was accounted for using a Bonferroni-adjusted phenome-wide significance threshold ($P < 2.6 \times 10^{-5}$). The red line denotes the Bonferroni-adjusted threshold, and the blue line denotes the nominally significant threshold ($P < 0.05$). The y-axis denotes the $\log_{10}(p\text{-value})$, and the x-axis denotes the disease category. Upright and inverted triangles denote positive and negative direction association with triangles colour-coded by disease category. Source data are provided as a Source Data file.

strong genetic and phenotypic interdependency. Across 200 loci, 177 were associated with >1 phenotype. These loci were associated with both highly correlated and weakly correlated CMR phenotypes, suggesting shared cardiac mechanisms and broader systemic influences, with some highly pleiotropic loci such as *ATXN2/SH2B3* showing extensive associations across multiple disease categories. Some pleiotropic associations indicate common pathways influencing CMR and CVD traits. For example, the *TBX3* (T-box transcription factor 3) locus for LVM, LVSV and RV LV ratio was associated with several cardiovascular diseases (e.g. HF, hypertension and ischaemic stroke). *TBX3* had a prioritisation score of 3 with an indicated regulatory role in congenital heart disease and cardiac remodelling[31] in zebrafish models.

On investigating the genetic overlap between LV and RV phenotypes and HF, we used summary data from Levin et al. for all-cause HF[22]. We found 23 colocalised loci with all-cause HF that shared the same causal variants (H4 > 0.8) with nearby genes that have supporting functional studies. For example, *ADAMTSL3* (A Disintegrin-Like And Metalloprotease Domain With Thrombospondin Type I Motifs-Like 3) associated with seven LV traits and RV-LV ratio, colocalised with HF, and had mouse model knockouts with cardiac dysfunction[32]. In boxer dogs, homozygous *STRN* (striatin), which colocalised with HF and was associated with LVMVR, was linked with the development of dilated cardiomyopathy[33]. *CAMK2D* (Calcium/Calmodulin Dependent Protein Kinase II Delta), which colocalised with HF and associated with LVMVR, RVESV, and RV LV ratio, was also identified as a potential HF therapeutic target by Rasooly et al. via Mendelian Randomisation[18].

Our systematic gene prioritisation identified 488 genes with prioritisation scores between 1-4, which underwent enrichment analysis. The top prioritised genes, *BAG3* and *ALPK3*, scored at 4, are well-established cardiomyopathy genes and known CMR loci. The 28 genes scored 3, contained 5 genes in novel CMR loci (*CASZ1, SLC4A7, ENG, HMCN2* and *ACTA2*), and highlighted potentially novel genes with multi-layered supporting evidence. 14 of these 28 genes were significantly enriched for cardiovascular knockout mouse phenotypes or cardiovascular pathways. For example, *HSPB7* within a known CMR locus encodes a heat shock protein, also colocalised with HF (H4 > 0.8). Chopra et al. reported restoration of cardiac contractile function in titin cardiomyopathy by knocking down *HSPB7*[34]. This result was also aligned with another HF-colocalised candidate gene, *YWHAE* (tyrosine 3-monooxygenase/tryptophan 5-monooxygenase activation protein). Chopra et al. identified *YWHAE* as being involved in cardiomyocyte structure and function, with its knockout in human stem cell-derived cardiomyocytes showing a similar functional profile to that of *TTN* mutation[34].

We performed an in-depth examination of druggability for 488 prioritised genes. Twenty-one interacted with cardiovascular drugs in Open Targets, 15 of which were significantly enriched in cardiovascular pathways and mouse model phenotypes. These 21 genes also included 15 genes that interact with an HF indicated drug (e.g. *MYH6/MYH7* interacting with Omecamtiv Mecarbil). Sixteen out of 21 genes are within previously reported CMR loci, including *NOS3* interacting with phosphodiesterase inhibitors such as Sildenafil[10]. *IGF1R* (Insulin-like growth factor 1 receptor), in a known CMR locus, was significantly enriched for cardiovascular pathways (e.g. heart development, circulatory system development and cardiac chamber development). *IGF1R* interacts with cardiovascular-related drugs, e.g. thrombin and metformin, and mouse studies have shown IGF1R signalling regulates cardiac fibrosis[35] and that IGF1R inhibition is a potential cardiovascular drug target[36]. Genes harboured by novel CMR loci have interactions with HF drugs, such as *GUCY1A3/GUCY1A1,* which interacts with Vericiguat[37].

Our drug mechanism enrichment analyses using DrugEnrichr also showed concordance between prioritised genes and established cardiovascular drug classes. Potassium-channel blockers and phosphodiesterase inhibitors were significantly enriched in our gene-drug

interactions. The drugs interacted with canonical targets such as *KCNH2* (anti-arrhythmic drugs), *PDE3A/B* (inotropes/vasodilators), *ITGB3* (antiplatelet agents) and *ATP1B2* (cardiac glycosides), aligning with our functional annotation-based gene prioritisation. Among the drugs targeting *PDE3* genes, anagrelide, which inhibits *PDE3A/B*, has been associated with reversible non-ischaemic cardiomyopathy[38]. *PDE3A/B* are both targets of several known CVD drugs (e.g. dipyridamole, enoximone and pentoxifylline). For *PDE3A* specifically, we found several lines of evidence supporting a potential cardiac-specific influence (ranging from variants with adverse effect directions in the LVMVR and LVMCF GWASs, significant enrichment for GTEx LV and coronary artery tissue upregulation, and Hi-C overlap in the RV). Polidovitch et al. showed that *PDE3A*, but not *PDE3B*, drives adverse ventricular remodelling under chronic pressure overload, with *PDE3A* upregulation promoting hypertrophy, dilation, fibrosis, and contractile dysfunction[39]. In contrast, Chung et al. found that *PDE3B* disruption, but not *PDE3A* disruption, protects against ischaemia/reperfusion injury via enhanced cAMP/PKA signalling[40]. Taken together, these studies suggest that *PDE3A* and *PDE3B* act on distinct biological mechanisms−*PDE3A* within sarcoplasmic reticular complexes impacting cardiac contractility and growth responses[39], and *PDE3B* within mitochondrial and metabolic signalling mechanisms relevant to acute ischaemic stress[40]. These results suggest HF treatment could benefit from selective PDE3A inhibition−improving on non-selective PDE (PDE3A/3B) inhibitors, such as milrinone, which are associated with poor HF outcomes with long-term use[41]. Further functional research is still needed to validate this adverse influence on cardiac tissues. However, overall, these findings reinforce established therapeutic mechanisms and suggest genetic targets that can improve our understanding of HF mechanisms and treatments.

We also identified *ENG* (endoglin) in a novel CMR locus, colocalised with HF (H4 > 0.8), and was also enriched in the "cardiac muscle tissue development" Gene Ontology term (adjusted *P*-valued $2.94 \times 10^{-15}$). Kapur et al. reported that reduced *ENG* expression limits cardiac fibrosis and improves HF survival[42]. Furthermore, elevated serum levels of soluble ENG have been associated with increased left ventricular filling pressure and more severe heart failure symptoms[43]. ENG is inhibited by carotuximab, an oncology medication, which was shown to prevent aortic endothelial cell dysfunction[44]. However, it should be noted that carotuximab is still undergoing a Phase 3 clinical evaluation for angiosarcoma, with no cardioprotective or cardiotoxicity effects reported to date.

Our study provided insights by conducting rare variant collapsing analyses, identifying significant associations for 13 genes, including cardiomyopathy genes (*TTN, RBM20, FHOD3, CSRP3* and *FLNC*) and others (*CORO6*[45], *DUSP13*[46], *GAA*[47], *GIP*[48], *CYP3A4*[49] and *SIRT4*[50]) with reported cardiovascular roles. For example, a murine study found that *SIRT4* promotes cardiac hypertrophy, fibrosis and functional impairment, and its inhibition could mitigate pathological cardiac hypertrophy[50]. Notably, *GIP* (gastric inhibitory polypeptide−known to stimulate insulin and glucagon secretion) has shown promise as a drug target for obesity[48], diabetes[51] and HF with preserved ejection fraction[52]. It also has an established role in glucose homeostasis, impacts cardiovascular signalling in mice[53] and has been suggested as a novel target for HF and diabetes[54].

The PheWAS of LV and RV PRSs illuminated relationships between the genetics of CMR traits and cardiovascular conditions. The PheWAS for LVEDV, LVESV, LVM and LVMVR all showed significant positive associations with HF, hypertension, cardiomyopathies, and cardiac dysrhythmias, aligning with epidemiological observations. Contrastingly, LVEF, LVMCF and LVGFI PRSs had strong negative associations with the same phenotypes, matching prior clinical knowledge. When focusing on non-cardiovascular phenotypes, the PRSs for LVEDV, LVSV and LVM BSA had significant positive associations with osteoarthritis and rheumatoid arthritis. Prior studies have shown a higher prevalence

of diastolic dysfunction and heart failure in individuals with osteoarthritis[55] and diastolic dysfunction, increased LVM and a higher risk of HF in individuals with rheumatoid arthritis[56,57], corroborating our PheWAS. Furthermore, multiple RV traits are associated negatively with mental health disorders. These findings highlight complex genetic interplay between cardiac structure and remodelling and various organs and systems.

This study benefited from several advantages, e.g. having the largest CMR GWAS sample size to date, including rare variant analysis for the first time, using the dense TOPMed imputation panel, extensive bioinformatic analysis and CMR phenotypes being the more precise and reproducible method to measure cardiovascular structure and systolic function. This study also has limitations, including being restricted to individuals of European ancestry and the sample size, although substantial, may be lacking in power to fully capture the genetic architecture, highlighted by modest proportions of variance explained by our lead variants. Further work is needed in larger cohorts to replicate the reported loci. Functional studies and longitudinal outcome studies are required to fully evaluate the biological roles of prioritised genes and potential drug targets in cardiac remodelling and HF. Additionally, several prioritised genes are expressed in multiple tissues, so off-target effects and systemic toxicity represent major translational challenges. Cardiac-specific functional validation, tissue-restricted targeting strategies and careful safety profiling will be essential before therapeutic conclusions can be drawn. Also, our GWAS is influenced by pleiotropic effects across the multiple traits studied, and advanced methods such as mt-COJO and structural equation modelling are necessary to robustly disentangle these effects.

In conclusion, this GWAS of 20 clinically relevant CMR LV and RV phenotypes identifies 200 associated loci, of which 58 were novel CMR loci. The colocalisation and shared causality with HF loci, in combination with cardiovascular pathway enrichment and potential therapeutic implications, underscore the translational relevance of these loci. Overall, these findings significantly enhance our understanding of the genetic architectures of LV and RV phenotypes. Our results highlight candidate genes and pathways underpinning cardiac structure and remodelling and create additional evidence for novel HF drug targets.

## Methods

### Ethical Compliance
Ethics approval for the UK Biobank study was obtained from the North West Centre for Research Ethics Committee (11/NW/0382). Informed consent was obtained from all participants in accordance with the UK Biobank ethics framework (https://www.ukbiobank.ac.uk/learn-more-about-uk-biobank/governance/ethics-advisory-committee). The study was designed and conducted in compliance with all applicable regulations for research involving human participants and adhered to the principles of the Declaration of Helsinki. The UK Biobank data was accessed by the study application 2964.

### UK Biobank
The UK Biobank is a population study of 502,664 participants aged between 49-60 years at the time of recruitment. The UK Biobank study's protocol[58] included comprehensive health and demographic questionnaires, clinic measurements, and participant blood samples taken. In total, 468,541 participants were genotyped and have WES data available. In accordance with standard UK Biobank policy, participants do not receive compensation. Genotyping was performed using UK Biobank Axiom arrays, with quality control (QC) and genome-wide imputation using the Haplotype Reference Consortium panel conducted by the UK Biobank team[59]. In this study, we use genotype data and cardiac phenotypes (derived from CMR) from the UK Biobank for 56,509 individuals (27,277 male and 29,232 female, mean age = 64.9 years).

### CMR left and right ventricular phenotypes
The analysis focused on eight LV and five RV CMR-derived phenotypes that quantify complementary aspects of ventricular structure, systolic function, myocardial remodelling, and integrated pump performance. LV structural measures included end-diastolic volume (LVEDV), reflecting chamber size at maximal filling; end-systolic volume (LVESV), reflecting residual volume after contraction; stroke volume (LVSV), representing the volume of blood ejected per beat; and LV mass (LVM), reflecting myocardial hypertrophy and remodelling. LV systolic function was assessed using LV ejection fraction (LVEF), the proportion of blood ejected per beat. Myocardial remodelling was further captured by the LV mass-to-volume ratio (LVMVR), a marker of concentric remodelling. Integrated LV pump performance was assessed using the LV global function index (LVGFI), which relates stroke volume to total LV volume and mass, and the LV myocardial contraction fraction (LVMCF), which reflects myocardial shortening relative to myocardial volume.

RV phenotypes included RV end-diastolic volume (RVEDV), RV end-systolic volume (RVESV), RV stroke volume (RVSV), RV ejection fraction (RVEF) and the RV/LV volume ratio, reflecting relative ventricular balance and coupling. To account for inter-individual differences in body size, selected volumetric and mass phenotypes are indexed to BSA (LVEDV, LVESV, LVSV, LVM, RVEDV, RVESV and RVSV), whereas ratio-based and functional measures are inherently size-normalised and therefore reported without additional BSA adjustment. Together, these 20 phenotypes were selected a priori to capture complementary and physiologically meaningful domains of ventricular size, contractile performance, myocardial remodelling, and pump efficiency across both ventricles, rather than to represent independent traits.

Prior to genetic association testing, all phenotypes were adjusted for key demographic, technical, and population structure covariates using linear regression. Specifically, each phenotype was regressed on age at imaging, sex (self-reported at the time of recruitment into the UK Biobank), genotyping array, imaging centre, and the first 10 genetic principal components to account for ancestry. When phenotypes were not indexed to BSA, height and weight were additionally included as covariates and regressed out. The residuals from these models were then inverse normal transformed to ensure normally distributed traits and reduce the impact of outliers. These covariate-adjusted, inverse normal-transformed phenotypes were used as the primary inputs for all downstream GWAS analyses. This pre-adjustment strategy ensured robust control for major confounders affecting ventricular structure and function, including body size–related effects already accounted for via BSA indexing or height and weight adjustment in the regression models as appropriate. Participant demographic information, principal components of ancestry, and ethnicity were further incorporated via both phenotype adjustment and sample filtering to enhance robustness.

### Genotype quality control
The genotype data underwent several filters for QC. Linkage disequilibrium pruning was performed using PLINK 2.0, a window size of 1000 kb, a step size of 50 SNPs, and an LD threshold of $r^2 < 0.4$. Following pruning, variants were filtered by a MAF threshold of 0.01, a minimum allele count (MAC) of 100, a genotype missingness cutoff of 0.1, a Hardy–Weinberg equilibrium $P$-value threshold of $1 \times 10^{-15}$, and an individual missingness threshold of 0.1.

### Genome-wide association study
In this study, we leverage densely imputed genotypes and whole-exome sequencing data along with multi-trait and colocalisation analyses to identify novel susceptibility loci and utilise a multilayered bioinformatics approach to identify drug re-purposing opportunities. GWAS was performed using whole-genome regression via REGENIE

(v3.3) in the UK Biobank Research Analysis Platform (RAP), providing single-trait association results for each of the 20 LV and RV image-derived phenotypes. All GWAS analyses were conducted using covariate-adjusted, inverse normal-transformed phenotypes, as described above. Genotypes were imputed using the TOPmed[60] imputation panel, and GWAS association testing was carried out on the resulting imputed genotype dosages (using REGENIE Step 2). TOPmed contains 487,180 individuals and approximately 400 million genetic variants. REGENIE step 2 analysis included setting minMAC (minimum minor allele count) of 20 to avoid testing extremely low-count variants. Following association testing, variants were filtered for downstream analyses based on minor allele frequency (MAF > 0.01) and imputation quality (INFO > 0.3), resulting in approximately 8,899,000 variants included in the final analyses. GWAS results from analysing 8,899,000 TOPmed variants (using Manhattan, QC, QQ, and regional plots) were visualised in R (v4.1.1) using packages ggplot (v3.4.2) and topr (v1.1.10). The TOPmed imputation panel was downsampled to 5000 randomly selected individuals in order to identify SNPs in LD. Loci were identified by SNPs within 1 Mb of lead variants ($P$-value $< 5 \times 10^{-8}$). Locus blocks were identified by numbered groups that grouped lead SNPs within 1 Mb of each other, and if they had LD > 0.4. To identify novel CMR loci, the loci were filtered against previously reported CMR loci by identifying lead variants within 1 Mb of previously reported CMR loci and identifying if any lead variants were in LD with the previously reported loci ($r^2 > 0.1$ and 4 Mb window). Previously reported CMR loci were manually curated from literature review and from programmatically identifying all LV and RV CMR studies from the GWAS catalogue (2024-06-17 release).

Genome-wide complex trait analysis (GCTA) version 1.94.1 was user to perform conditional and joint (COJO) analysis to identify independent signals per loci within each chromosome. The analysis was run with setting --cojo-slct, which applies stepwise model selection to select the independent variants by fitting all SNPs simultaneously and conditioning on each other.

### Rare variant analysis

Rare variant analysis was performed using REGENIE (v3.3). REGENIE step 2 was used for gene-burden testing using SKAT-O and ACATO-full tests, with a variant effect predictor (VEP) annotation mask (using all of the masks and taking the global $P$-value), and variants filtered by MAF < 0.01, and a minMAC of 1. Conditional analysis was performed on rare variant associations with nearby common significant SNPs (MAF > 0.01) in REGENIE. For the VEP annotation masks, we applied six functional masks to capture different predicted deleterious effects: loss-of-function variants (mask 1), high-confidence loss-of-function variants (mask 2), loss-of-function plus missense variants with REVEL score >0.5 (mask 3), high-confidence loss-of-function plus missense variants with REVEL > 0.5 (mask 4), loss-of-function plus all missense variants (mask 5), and high-confidence loss-of-function plus all missense variants (mask 6). Gene-burden testing was performed across all masks, and the global $P$-value (taking into account all 6 masks) was reported for each gene. Conditional analyses were performed to evaluate whether rare variant associations were independent of nearby common variants (MAF > 0.01).

### Heritability, genetic correlation, and multi-trait analysis

Heritability estimates were inferred by the restricted maximum likelihood method in GCTA, and genetic correlation by LD score regression (LDSC). GCTA was used to estimate the heritability, and LDSC was used to calculate genetic correlation for each phenotype—using LDSC v1.0.1 and Python v2.7. The reference LD scores and weights were derived from the pan-UK Biobank European ancestry dataset[61]. Phenotype pairs with a correlation >0.7 were selected for multi-trait analysis using multi-trait analysis of GWAS MTAG (v1.0.8). To avoid redundancy and artificial inflation arising from the same phenotype

being analysed with and without BSA indexing, pairs consisting of a phenotype and its BSA-adjusted counterpart (e.g. LVEDV and LVEDV BSA) were explicitly excluded from multi-trait analyses, even when these exceeded the >0.7 correlation threshold. This ensured that each multi-trait analysis combined biologically related but non-duplicative ventricular phenotypes. As a result, only LVEDV, LVEDV BSA and RVEDV were included in more than one multi-trait pairing, each time with a distinct, non-rescaled correlated phenotype. Resulting GWAS summary data from MTAG were filtered by MAF > 0.01 and INFO > 0.3.

To account for the number of tests across all phenotypes with summary statistics, the Galwey method was used to calculate an adjusted $P$-value threshold for LV and RV phenotypes ($P < 7.28 \times 10^{-9}$ in comparison to the conventional $5 \times 10^{-8}$ threshold), using the poolr R package (v1.2-0). The Galwey estimates the effective number of independent phenotypes based on the eigenvalues of the phenotype correlation matrix. This eigenvalue-based approach refines earlier methods (e.g. Nyholt, Li and Ji) and provides a more accurate adjustment by accounting for phenotype interdependence, yielding an effective number of 6.85 tests[62].

### Functional annotation of variants

Lead variants and their proxies ($r^2 < 0.01$) were filtered to 99% credible sets, using the Bayesian method as previously described by Wakefield[63]. All variants in the 99% credible sets were interrogated with VEP (108 release) to describe their locations and predicted function using several risk prediction tools, including SIFT, PolyPhen-2 and REVEL. The non-coding variants were annotated with CADD (v1.7) and RegulomeDB (v2.2). Damaging variants were identified via CADD > 20 or RegulomeDB rank ≤1f, and damaging prediction in either SIFT or PolyPhen. All variants were also analysed with DEPICT. Colocalisation analysis of GWAS and cis-expression quantitative trait loci (cis-eQTL) signals from cardiovascular tissues, including the aorta, coronary artery, tibial artery, left atrial appendage and left ventricle, was conducted using data from the GTEx project version 8 and coloc (v5.2.3). We also ran colocalisation of our GWAS against a GWAS performed by Levin et al. for all-cause heart failure (GWAS Catalogue accession ID: GCST90162626)[22].

### Cross-referencing left and right ventricular loci in other CMR traits

We looked up our lead variants in the summary data available from recently published genome-wide association studies of CMR-derived LV and RV phenotypes with comparable sample size, using curated CMR studies (DOI references included in Supplementary Data 3) and the GWAS catalogue (2024-06-17 release). We also used the PhenoScanner27 database v2 (http://www.phenoscanner.medschl.cam.ac.uk/) and the GWAS catalogue via the gwasrapidd R package (v0.99.14) to cross-reference our lead variants and their nearby SNPs (in LD $r^2 \geq 0.8$ and in the 99% credible sets) with the GWAS of other traits.

### Gene prioritisation and druggability analysis

All genes were annotated to the 200 loci within a 10 kb window, identifying 1339 genes in total. The genes were then first annotated with PoPS (v0.2), using MAGMA (v1.10). PoPS utilises gene-level association statistics from GWAS summary data and analyses polygenic enrichment of gene features from cell-type-specific gene expression, biological pathways and protein–protein interactions[23]. We then utilised multiple bioinformatic resources and prioritisation tools to provide a stronger evidence base for prioritisation. The genes underwent systematic scoring, assigning each gene a score between 1 and 4, with 4 being the highest score and 1 being the lowest. The scoring weighted the 1339 genes across several lines of evidence (PoPS, colocalisation with HF, OMIM and ClinGen cardiovascular disease annotation, mouse model cardiovascular phenotype annotation, Hi–C annotation, and

Article

GTEx cardiovascular annotation). Based on applying equal weighting to each of these criteria, the genes were scored from 1 to 4 (lowest to highest, with genes with a score of four having the highest prioritisation and having all annotations, and a gene with a score of one only having one of these annotations. Genes were assigned these scores based on having the highest PoPS per locus, OMIM and ClinGen cardiovascular disease annotation, GTEx cardiovascular tissue enrichment or presence of Hi-C data for cardiovascular tissues, and colocalisation >0.8 H4 with the Levin et al. HF GWAS. For example, if a gene had all four of these annotation groups, it was assigned a score of four, and if it only had one annotation, it was assigned a score of one. Excluding PoPS and colocalisation analysis, all annotations were collected via Enrichr (v3.2)[64], ClinGen (accessed: 31-01-2024)[65] and IMPC mouse model data (v21.0), and promoter capture Hi-C data from Jung et al.[66].

Further annotations were collected including Exomiser scores[67] combined for LV phenotypes (HP:0005162, HP:0001711, HP:0025169, HP:0034314, HP:0012664, HP:0033754, HP:0001712, HP:0031482, HP:0034385, HP:0012666, HP:0012665, HP:0012663, HP:0011664, HP:0025168, HP:4000141, HP:0033755, and HP:00337560) and RV phenotypes (HP:0001707, HP:0033118, HP:0012816, HP:0001667, HP:0001708, HP:0005133 and HP:0011663), pathway enrichment using g:Profiler, druggability likelihood predicted by DrugnomeAI[68], gene–drug interactions and drug warnings from Open Targets (v24.03), gene–drug interactions and druggability categories from DGIdb (June 2023 release).

Druggability analysis annotated genes to their drug interactions as denoted in Open Targets and DGIdb. Selected groups of scored genes were identified by whether their interacting drugs had cardiovascular disease targets or not, as denoted in Open Targets and cross-referenced by an expert clinician. Genes that had no drug interactions in either Open Targets or DGIdb but were annotated in DGIdb as "druggable" underwent further analysis investigating their DrugnomeAI predicted druggability probability and their cardiovascular pathway and mouse model enrichment (via gprofiler and IMPC). Enrichment analysis for drug mechanisms of action was performed via the DrugEnrichr web interface[64].

### Phenome-wide association studies
PheWAS was conducted using PRS-CS-auto[30] for each phenotype's summary statistics. The PheWAS phenotypes were derived from ICD-10 codes, updated on 13 March 2024. In total, 1855 phenotypes were defined in the phecode system. PheWAS was performed using the PheWAS R package (v0.99).

### Reporting summary
Further information on research design is available in the Nature Portfolio Reporting Summary linked to this article.

### Data availability
The full genome-wide association summary statistics generated in this study have been deposited in the GWAS Catalogue under accession codes GCST90797570–GCST90797613 [https://www.ebi.ac.uk/gwas/downloads/summary-statistics]. These summary statistics are fully available for download and use. The raw individual-level genotype and cardiac MRI data from the UK Biobank are protected and are not publicly available due to data privacy laws; access can be obtained through application to the UK Biobank Access Management System (https://www.ukbiobank.ac.uk/enable-your-research/apply-for-access), subject to UK Biobank approval. The data generated in this study for all figures and source data are provided in the Supplementary Information/Source Data file. Source data are provided with this paper. The HF GWAS performed by Levin et al. and used in our colocalization analysis is available in the GWAS Catalogue (accession ID: GCST90162626). Source data are provided with this paper.

### Code availability
Publicly available software tools were used to perform all analyses. PLINK, REGENIE, GCTA, LDSC, MTAG, DEPICT, MAGMA, PoPS, DrugnomeAI, Exomiser, g:Profiler, Enrichr, Drugenrichr, coloc, PheWAS. Code for performing the GWAS analysis can be found here: https://github.com/hlnicholls/CMR_GWAS and https://zenodo.org/records/18431669[69]. The algorithms for CMR image analysis are available in https://github.com/baiwenjia/ukbb_cardiac. For automated CMR image analysis, we used Python v3.6 and TensorFlow v1.9.0. Manual analysis of CMR studies was performed using Cvi42 software (v5.1.1) (https://www.circlecvi.com).

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

## Acknowledgements

N.A. and H.L.N. acknowledge the funding support from the Medical Research Council (MR/X020924/1). M.M.S. recognises his British Heart Foundation's (BHF) Clinical Research Training Fellowship (FS/CRTF/22/24353). We acknowledge the British Heart Foundation for funding the manual analysis to create a CMR imaging reference standard for the UK Biobank imaging resource in 5000 CMR scans (PG/14/89/31194; S.E.P.). This work acknowledges the support of the National Institute for Health and Care Research Barts Biomedical Research Centre (NIHR203330); a delivery partnership of Barts Health NHS Trust, Queen Mary University of London, St George's University Hospitals NHS Foundation Trust and St George's University of London (N.A., P.B.M. and S.E.P.). The UK Biobank was established by the Wellcome Trust medical charity, the Medical Research Council, the Department of Health, the Scottish Government and the Northwest Regional Development Agency. It has also received funding from the Welsh Assembly Government and the British Heart Foundation. Barts Charity (G-002346) contributed to the fees required to access UK Biobank data [access application #2964]. This research was conducted using the UK Biobank Resource under application 2964. We thank all UK Biobank participants and staff.

## Author contributions

H.L.N. and N.A. contributed to the study design, data analysis, and interpretation, with N.A. providing supervision. M.M.S. contributed to paper review and editing. These authors contributed equally to the paper review: J.D.V., H.A., C.A.A.C., M.Y.K., S.E.P. and P.B.M.

## Competing interests

S.E.P. provides consultancy to Circle Cardiovascular Imaging, Inc., Calgary, Alberta, Canada. The remaining authors declare no competing interests.
