## [Transparent Peer Review file · Nature Communications]

Genome-wide analysis of cardiac ventricular phenotypes reveals novel loci and therapeutic targets for heart failure

Corresponding Author: Dr Nay Aung

Version 0:

Reviewer comments:

Reviewer #2

(Remarks to the Author)

The authors present a revised version of their manuscript evaluating the association between common- and rare- genetic variants and cardiac MRI measures among UK Biobank participants. The authors have revised their drug-repurposing/repositioning analyses, updated the pleiotropy analysis to consider shared signals between uncorrelated phenotypes, and improved the discussion. A few additional comments:

- 1) The circular Manhattan plots in Figure 2 are largely uninterpretable given the large number of tracks/signals. The Miami plots in Supplemental Figure 2 provide a more interpretable representation of the results – the authors may want to consider this for a main figure instead.
- 2) In the Discussion the authors conclude that the high degree of genetic correlation and degree of heritability imply these traits are highly polygenic. Without formal polygenicity quantification, the authors may consider removing this element and focusing more on pleiotropic signals and genetic correlation.
- 3) Please clarify the GWAS methods – the authors describe using variants with a minor allele frequency of > 0.01 , but earlier in the section describe setting a minimum minor allele count of 20 in Step 2 of REGENIE, which represents a much lower allele frequency. Based on the total number of variants tested (~9 million), I suspect this corresponds to $MAF > 0.01$ rather than $MAC > 20$.

Reviewer #3

(Remarks to the Author)

This manuscript reports an extensive genome-wide investigation of 20 CMR-derived ventricular phenotypes in over 56,000 UK Biobank participants. The authors integrate common variant GWAS, rare variant burden analyses, HF colocalization, gene prioritization, pathway enrichment, polygenic risk scoring, and druggability evaluation. The scale of the dataset and the breadth of analyses are impressive, and the work provides meaningful insights into ventricular remodeling and heart failure biology. The revisions to the druggability section improve clarity and appropriately limit overinterpretation. However, several important issues remain.

- A major concern is that the manuscript uses as many as 20 CMR traits without sufficient justification. Many of these measures are physiologically interdependent, which makes it difficult to interpret pleiotropic associations and may exaggerate the number of loci appearing to influence multiple traits.
- The manuscript also does not sufficiently describe the definitions and physiological meaning of the CMR phenotypes. Readers without a cardiovascular imaging background will find it difficult to interpret the biological relevance of these associations.
- CMR is not the most widely used method to evaluate LV function and the appeal of many of these parameters will be of limited value and interest to readers where the terminology is esoteric (even to cardiologists) who routinely evaluate patients
- The handling of important confounders is not well explained. Ventricular structure and function are strongly affected by body size and hemodynamic factors, yet the GWAS models do not make it clear how these variables were included or

adjusted for. More detail on covariates, normalization, and modeling choices is needed to assess the robustness of the findings.

- Druggability portion seems not to be of interest since many of these genes are expressed in other tissue and off target unintended effects would be of major concern.

Reviewer #4

(Remarks to the Author)

This paper represents the largest and most comprehensive GWAS analyses of MRI based imaging measures of heart function. It is 2x the size of a previous paper from the same authors that used 29000 UK Biobank participants in its discovery step. Convincing new loci are reported.

Main points

1. How should the reader interpret the signals that come up with the multi trait analyses ? what do these mean if they do not come up with a single trait ? and what do they mean if they do come up in a single trait but are or similar or stronger significance ? as the traits are tightly correlated with each other it is not clear. This is especially important when 2 GWAS could be on the same trait but one with the body surface area correction.
2. Related to point 1, there are a lot of traits and a lot of multi trait analyses. Would it make sense to limit the main results to the traits adjusted for body surface and then just talk about the traits not adjusted for body surface in the supplementary information as sensitivity analyses ? this will make it easier to keep track of the main findings. I suspect there are very few signals that would be "lost".

Minor points

1. It would help readability to start sentences with written numbers e.g. Twenty one rather than 21.
2. Main results. There are some slightly confusing statements scattered throughout the manuscript, for example line 103 the sentence starting "Using single trait" GWAS I find confusing, although I think I can work it out, but perhaps expand to explain more clearly. In the abstract "rare variant analyses found significant burden..." is not clear .
3. Line 220 and paragraph. What is meant by "interacted" in this context ? I presume this is about the gene product encoded by the gene, not genetic variants ?
4. Results rare variant analyses. Line 257 worth stating more clearly which rare variants went in, not just by frequency.
5. Methods. What do the authors mean by "indexed on" ? when referring to the body surface area correction ?
6. Worth spelling out minMAC first time used in methods
7. Lines 472 and the use of TOPMED section. This section of the methods is about the imputation panel ? GWAS was not "performed" in TOPMED ? check wording.
8. The circular manhattan plots look nice but are a little hard to read. It is not clear how the gene names map to the traits, and a colour system is challenging. Perhaps better to simply leave to the supplementary figures in straight lines to aid readability but taking up more room ? ST1 the Genetic correlations maybe better as a main figure ?
9. From page 26 there is a dump of data that is unreadable in the pdf version I have. Has the format been disrupted ?
10. See point 5 but the SF manhattens are hard to read , consider placing one on each row not two and using more distinct colours.
11. SF3 what does "interacting associations" mean ?

Version 1:

Reviewer comments:

Reviewer #2

(Remarks to the Author)

The authors have addressed my comments.

Reviewer #3

(Remarks to the Author)

The authors have adequately addressed my previous concerns by clarifying the rationale for the inclusion of multiple correlated CMR phenotypes, expanding the physiological definitions of imaging traits, and providing a transparent description of covariate adjustment.

One minor issue remains regarding the readability of several tables and supplementary tables, which appear to have formatting issues following conversion from Excel to PDF. These should be revised.

Reviewer #4

(Remarks to the Author)

Many thanks to the authors for addressing the questions. There is still a formatting issue in the main table in my version but I will leave that with the editors and authors.

REVIEWER COMMENTS

Reviewer #2 (Remarks to the Author):

The authors present a revised version of their manuscript evaluating the association between common- and rare- genetic variants and cardiac MRI measures among UK Biobank participants. The authors have revised their drug-repurposing/repositioning analyses, updated the pleiotropy analysis to consider shared signals between uncorrelated phenotypes, and improved the discussion.

A few additional comments:

1) The circular Manhattan plots in Figure 2 are largely uninterpretable given the large number of tracks/signals. The Miami plots in Supplemental Figure 2 provide a more interpretable representation of the results – the authors may want to consider this for a main figure instead.

We thank the reviewer for their feedback. We agree that the circular Manhattan plots are visually dense. We have replaced Figure 2 in the main manuscript with the Miami Manhattan plots (previously shown in Supplementary Figure 2), updated for readability (adjusting colour, font size, and label positioning).

2) In the Discussion the authors conclude that the high degree of genetic correlation and degree of heritability imply these traits are highly polygenic. Without formal polygenicity quantification, the authors may consider removing this element and focusing more on pleiotropic signals and genetic correlation.

To avoid overinterpretation, we have revised the Discussion to remove the explicit claim that these traits are highly polygenic and now focus instead on the extensive pleiotropy and shared genetic architecture reflected by the strong genetic correlations and overlapping loci.

We have reworded the Discussion sentence to:

“Heritability estimates for LV and RV traits ranged from 18.3% to 35.8%, alongside moderate to high genetic correlation, reflecting strong genetic and phenotypic interdependency.”

3) Please clarify the GWAS methods – the authors describe using variants with a minor allele frequency of > 0.01 , but earlier in the section describe setting a minimum minor allele count of 20 in Step 2 of REGENIE, which represents a much lower allele frequency. Based on the total number of variants tested (~9 million), I suspect this corresponds to $MAF > 0.01$ rather than $MAC > 20$.

We thank the reviewer for highlighting this ambiguity. We agree that the description of the minor allele frequency and minor allele count thresholds required clarification. In REGENIE step 2, we applied a minimum minor allele count (MAC) threshold of 20 as part

of the association testing framework. However, variants were subsequently filtered for inclusion in downstream analyses using a minor allele frequency (MAF) threshold of >0.01 and imputation quality ($INFO > 0.3$), which determined the final set of ~ 8.9 million variants tested. We have now revised the Methods to clearly distinguish the MAC filter applied during REGENIE step 2 from the MAF-based filtering used downstream for defining the analysed variant set:

“REGENIE step 2 analysis included setting minMAC (minimum minor allele count) of 20 to avoid testing extremely low-count variants. Following association testing, variants were filtered for downstream analyses based on minor allele frequency ($MAF > 0.01$) and imputation quality ($INFO > 0.3$), resulting in approximately 8,899,000 variants included in the final analyses. GWAS results from analysing 8,899,000 TOPmed variants (using Manhattan, QC, QQ, and regional plots) were visualised in R (v4.1.1) using packages ggplot (v3.4.2) and topr (v1.1.10).”

Reviewer #3 (Remarks to the Author):

This manuscript reports an extensive genome-wide investigation of 20 CMR-derived ventricular phenotypes in over 56,000 UK Biobank participants. The authors integrate common variant GWAS, rare variant burden analyses, HF colocalization, gene prioritization, pathway enrichment, polygenic risk scoring, and druggability evaluation. The scale of the dataset and the breadth of analyses are impressive, and the work provides meaningful insights into ventricular remodeling and heart failure biology. The revisions to the druggability section improve clarity and appropriately limit overinterpretation. However, several important issues remain.

- A major concern is that the manuscript uses as many as 20 CMR traits without sufficient justification. Many of these measures are physiologically interdependent, which makes it difficult to interpret pleiotropic associations and may exaggerate the number of loci appearing to influence multiple traits.

We thank the reviewer for this important comment regarding the use of 20 CMR-derived traits and the potential impact of physiological interdependence on locus interpretation. We agree that many ventricular phenotypes are biologically correlated, reflecting shared aspects of cardiac structure, function, and remodelling. However, our intent in analysing a broad panel of CMR traits was precisely to capture this multidimensional and interdependent biology rather than to treat each trait as an isolated entity. Each phenotype was selected to reflect a physiologically distinct aspect of ventricular structure, function, or remodelling. By including both volumetric and functional measures, we can identify loci that differentially influence myocardial hypertrophy, contractility, or integrated pump performance.

Several steps were taken to address the challenges posed by interdependence. First, we formally quantified genotypic correlations between all traits using LD score regression and explicitly leveraged this structure in our multi-trait discovery framework, restricting

multi-trait analyses to phenotype pairs with high genotypic correlation ($r_g > 0.7$). Second, we applied an eigenvalue-based multiple testing correction (Galwey method) to account for the effective number of independent phenotypes, thereby reducing the risk of inflated significance due to correlated traits. Third, we explicitly distinguish between loci showing broad pleiotropic effects across multiple traits and those with greater trait specificity (e.g., identifying the number of phenotypes a locus associates with across all 20 phenotypes in Supplementary Table 3), and we interpret these patterns in the context of shared versus phenotype-specific cardiac biology. For example, gene-drug targets in Supplementary Table 14 were interpreted with considering their GWAS effect directions for interrelated phenotypes to identify potential targets of interest (e.g. *PDE3A* associations had lower LVMCF and higher LVMVR effect sizes, which may indicate an influence on adverse cardiac remodelling).

To further address this point, we have expanded our Methods section to justify our selected CMR phenotypes:

“Together, these 20 phenotypes were selected a priori to capture complementary and physiologically meaningful domains of ventricular size, contractile performance, myocardial remodelling, and pump efficiency across both ventricles, rather than to represent independent traits.”

- The manuscript also does not sufficiently describe the definitions and physiological meaning of the CMR phenotypes. Readers without a cardiovascular imaging background will find it difficult to interpret the biological relevance of these associations.

We fully agree that accessibility to readers without a cardiovascular imaging background is essential for appropriate biological interpretation. To address this, we have expanded our Introduction and Methods sections to explicitly define each LV and RV phenotype and to describe their physiological meanings, including measures of ventricular size (e.g., EDV, ESV), systolic performance (e.g., EF, SV), myocardial geometric remodelling (e.g., LVM, LVMVR), and integrated pump efficiency (e.g., LVGFI, LVMCF). Additionally, we have expanded our Supplementary Table 1 to include definitions per phenotype acronym, providing a clear reference for readers. We have also clarified the rationale for body surface area indexing and the complementary biological domains captured by the 20 traits. These additions now provide clear clinical and physiological context for all imaging-derived measures and directly support interpretation of the genetic associations.

We have expanded the Introduction here:

“Aberrations in cardiac ventricular structure and function underlie heart failure (HF), a leading cause of mortality and morbidity worldwide. Cardiac imaging phenotypes capture complementary aspects of cardiac anatomy and physiology, including chamber size, myocardial hypertrophy, and contractile performance, and are key measures used in early diagnosis of cardiac abnormalities. Left ventricular (LV) volumes represent the global size of the main pumping chamber, with end-diastolic volume reflecting preload and chamber dilatation, and end-systolic volume reflecting residual blood after contraction and systolic

efficiency. LV mass quantifies myocardial hypertrophy and remodelling, while LV stroke volume and LV ejection fraction (LVEF) represent pump function and systolic performance. Notably, left ventricular (LV) volume, LV mass, and LV systolic function measured by LV ejection fraction, are instrumental in HF diagnosis and prognostication. Recently, novel LV functional measures, LV global function index (LVGFI) and LV myocardial contraction fraction (LVMCF), have demonstrated superior predictive capabilities for adverse cardiovascular outcomes by integrating anatomic information in estimating global systolic performance. In parallel, right ventricular (RV) structure and function have independent and incremental roles in determining HF, with RV volumes, stroke volume, and ejection fraction reflecting pulmonary vascular loading, RV contractile function, and ventricular interdependence.”

We have also expanded our Methods section under “CMR Left and Right Ventricular Phenotypes”:

“The analysis focused on eight LV and five RV CMR-derived phenotypes that quantify complementary aspects of ventricular structure, systolic function, myocardial remodelling, and integrated pump performance. LV structural measures included end-diastolic volume (LVEDV), reflecting chamber size at maximal filling; end-systolic volume (LVESV), reflecting residual volume after contraction; stroke volume (LVSV), representing the volume of blood ejected per heart beat; and LV mass (LVM), reflecting myocardial hypertrophy and remodelling. LV systolic function was assessed using LV ejection fraction (LVEF), the proportion of blood ejected per heart beat. Myocardial remodelling was further captured by the LV mass-to-volume ratio (LVMVR), a marker of concentric remodelling. Integrated LV pump performance was assessed using the LV global function index (LVGFI), which relates stroke volume to total LV volume and mass, and the LV myocardial contraction fraction (LVMCF), which reflects myocardial shortening relative to myocardial volume.

RV phenotypes included RV end-diastolic volume (RVEDV), RV end-systolic volume (RVESV), RV stroke volume (RVSV), RV ejection fraction (RVEF), and the RV/LV volume ratio, reflecting relative ventricular balance and coupling. To account for inter-individual differences in body size, selected volumetric and mass phenotypes are indexed to body surface area (BSA) (LVEDV, LVESV, LVSV, LVM, RVEDV, RVESV, and RVSV), whereas ratio-based and functional measures are inherently size-normalised and therefore reported without BSA adjustment. Together, these 20 phenotypes were selected a priori to capture complementary and physiologically meaningful domains of ventricular size, contractile performance, myocardial remodelling, and pump efficiency across both ventricles, rather than to represent independent traits.”

- CMR is not the most widely used method to evaluate LV function and the appeal of

many of these parameters will be of limited value and interest to readers where the terminology is esoteric (even to cardiologists) who routinely evaluate patients

We thank the reviewer for this comment and acknowledge that CMR-derived parameters are not yet used routinely in all clinical settings. However, CMR is widely regarded as the reference standard for the quantitative assessment of ventricular volumes, mass, and systolic function due to its high spatial resolution (from superior signal-to-noise ratio), three-dimensional coverage, and absence of geometric assumptions. As such, it provides a uniquely precise and reproducible framework for genetic discovery, even when some derived metrics are not used in routine clinical care. We use CMR as a means to accurately measure cardiac structural and functional phenotypes in order to elucidate their relevance in heart failure, with insights that are generalisable beyond the CMR modality itself.

Importantly, while some parameters such as LVGFI and LVMCF may be less familiar in day-to-day clinical practice, they capture integrated aspects of cardiac pump performance and myocardial shortening that are not fully represented by conventional measures such as ejection fraction alone. Several large population-based and prognostic studies have demonstrated that these functional indices provide complementary information and improved sensitivity to subclinical ventricular dysfunction and remodelling^{1–5}. Their inclusion in this study was therefore motivated by biological relevance and sensitivity to early disease processes rather than by current clinical adoption alone.

To ensure accessibility for a broad readership, we have now expanded the Introduction and Methods sections to clearly define all CMR phenotypes and their physiological interpretation (as outlined in our response to the previous point), explicitly linking each metric to established concepts of ventricular size, contractile performance, myocardial remodelling, and pump efficiency. We believe this substantially improves interpretability for readers who do not routinely work with advanced CMR-derived parameters, while preserving the value of these quantitative traits for genetic discovery.

- The handling of important confounders is not well explained. Ventricular structure and function are strongly affected by body size and hemodynamic factors, yet the GWAS models do not make it clear how these variables were included or adjusted for. More detail on covariates, normalization, and modeling choices is needed to assess the robustness of the findings.

We thank the reviewer for highlighting the importance of clearly describing the handling of key confounders. We agree that ventricular structure and function are strongly influenced by body size, demographic factors, and technical variation. All CMR phenotypes were adjusted prior to GWAS by regressing each trait on age at imaging, sex, genotyping array, imaging centre, and the first 10 principal components of ancestry. We additionally included height and weight as covariates in the regression models when phenotypes were not indexed to BSA. The rationale for considering both BSA-indexed and unindexed phenotypes was explained in our response to Reviewer 4. This approach

ensures that the impact of body size is accounted for in our models. The residuals from these models were then inverse normal transformed and used as the phenotypes for all downstream GWAS analyses. We have now added a detailed description of this covariate adjustment and normalisation strategy to the Methods section to clarify the robustness of the modeling approach. With respect to haemodynamic factors such as blood pressure and heart rate, we elected to investigate their influence through pleiotropy analyses. This was intentional, as these factors may act as mediators, rather than confounders, of changes in cardiac structure and function.

“Prior to genetic association testing, all phenotypes were adjusted for key demographic, technical, and population structure covariates using linear regression. Specifically, each phenotype was regressed on age at imaging, sex, genotyping array, imaging centre, and the first 10 genetic principal components to account for ancestry. When phenotypes were not indexed to BSA, height and weight were additionally included as covariates and regressed out. The residuals from these models were then inverse normal transformed to ensure normally distributed traits and reduce the impact of outliers. These covariate-adjusted, Inverse normal transformed phenotypes were used as the primary inputs for all downstream GWAS analyses. This pre-adjustment strategy ensured robust control for major confounders affecting ventricular structure and function, including body size-related effects accounted for via BSA indexing or height and weight adjustment in the regression models as appropriate. Participant demographic information, principal components of ancestry, and ethnicity were further incorporated via both phenotype adjustment and sample filtering to enhance robustness.”

- Druggability portion seems not to be of interest since many of these genes are expressed in other tissue and off target unintended effects would be of major concern.

We fully agree that many of the prioritised genes are expressed in multiple tissues and that off-target and unintended systemic effects represent a major challenge for therapeutic translation. The purpose of the druggability analysis in this study was not to nominate immediate clinical candidates, but rather to provide a hypothesis-generating framework to highlight mechanistic links between ventricular remodelling genetics and existing pharmacological pathways. This aligns with recent studies identifying the importance of GWAS for drug development. Research has shown that in the past decade 63% of FDA approved drugs have had genetic support⁶, and that gene-drug targets with supporting evidence such as GWAS associations with supporting bioinformatic analysis are 2.6 times more likely to identify successful drug mechanisms⁷.

We have now clarified this intent in the Discussion and explicitly acknowledged the risks of non-cardiac expression, off-target toxicity, and the need for cardiac-specific functional validation before any translational conclusions can be drawn. Importantly, several highlighted targets (e.g., *PDE3A*, *CAMK2D*, *IGF1R*, and *ENG*) already have cardiac-relevant functional data supporting disease-specific effects, but we agree that tissue-specific targeting strategies and careful safety profiling will be essential for future therapeutic development.

We have added this as a dedicated limitation:

“Additionally, several prioritised genes are expressed in multiple tissues, so off-target effects and systemic toxicity represent major translational challenges. Cardiac-specific functional validation, tissue-restricted targeting strategies, and careful safety profiling will be essential before therapeutic conclusions can be drawn.”

Reviewer #4 (Remarks to the Author):

This paper represents the largest and most comprehensive GWAS analyses of MRI based imaging measures of heart function. It is 2x the size of a previous paper from the same authors that used 29000 UK Biobank participants in its discovery step. Convincing new loci are reported.

Main points

1. How should the reader interpret the signals that come up with the multi trait analyses ? what do these mean if they do not come up with a single trait ? and what do they mean if they do come up in a single trait but are or similar or stronger significance ? as the traits are tightly correlated with each other it is not clear. This is especially important when 2 GWAS could be on the same trait but one with the body surface area correction.

We thank the reviewer for their feedback. Signals identified in multi-trait analyses should be interpreted in the context of the genetic correlation among the traits. When a locus is identified in multi-trait analysis but not in any single-trait GWAS, this typically indicates that the variant exerts a shared, moderate effect across several correlated phenotypes, insufficient to reach genome-wide significance in isolation. Conversely, loci identified in both single- and multi-trait analyses, especially with similar or stronger significance in single-trait GWAS, likely represent variants with more specific or pronounced effects on that particular phenotype. In our study, the high correlation among LV and RV traits implies that many loci reflect underlying shared cardiac biology rather than trait-specific effects. Overall, multi-trait signals provide complementary insights by highlighting pleiotropic loci affecting multiple related aspects of ventricular structure and function, whereas single-trait signals reflect more specific or stronger effects for individual phenotypes.

However, we agree that particular care is required when applying multi-trait analysis to phenotypes defined with and without body surface area (BSA) correction. To avoid introducing artificial correlation or redundant signals, we explicitly excluded pairs consisting of the same phenotype with and without BSA adjustment (e.g. LVEDV and LVEDV BSA) from the multi-trait analyses, even when these pairs exceeded our genotypic correlation threshold ($r_g > 0.7$). We have now expanded the Methods section and updated the analysis flow diagram (Figure 1) to clearly describe this phenotype selection strategy.

In our Methods we have added:

“To avoid redundancy and artificial inflation arising from the same phenotype being analysed with and without BSA indexing, pairs consisting of a phenotype and its BSA-adjusted counterpart (e.g. LVEDV and LVEDV BSA) were explicitly excluded from multi-trait analyses, even when these exceeded the > 0.7 correlation threshold. This ensured that each multi-trait analysis combined biologically related but non-duplicative ventricular phenotypes. As a result, only LVEDV, LVEDV BSA, and RVEDV were included in more than one multi-trait pairing, each time with a distinct, non-rescaled correlated phenotype.”

We also updated Figure 1:

Figure 1. Flowchart of analysis for LV and RV GWAS. GWAS, genome-wide association study, CMR, cardiovascular magnetic resonance; LV, left ventricle; RV, right ventricle; RVEDV, right ventricular end-diastolic volume; RVESV, right ventricular end-systolic volume; RVSV, right ventricular stroke volume; RVEF, right ventricular ejection fraction; EA, effect allele; NEA, non-effect allele; SE, standard error; RVEDV, Right Ventricular End-Diastolic Volume; RVESV, Right Ventricular End-Systolic Volume; RVSV, Right Ventricular Stroke Volume; RVEF, Right Ventricular Ejection Fraction; RV LV ratio, Right Ventricular to Left Ventricular size ratio; RVEDV BSA, Right Ventricular End-Diastolic Volume indexed to Body Surface Area; RVESV BSA, Right Ventricular End-Systolic Volume indexed to Body Surface Area; RVSV BSA, Right Ventricular Stroke Volume indexed to Body Surface Area; LVEDV, Left Ventricular End-Diastolic Volume; LVESV, Left Ventricular End-Systolic Volume; LVEF, Left Ventricular Ejection Fraction; LVSV, Left Ventricular Stroke Volume; LVM, Left Ventricular Mass; LVMVR, Left Ventricular Mass to Volume Ratio; LVGFI, Left Ventricular Global Function Index; LVMCF,

Left Ventricular Myocardial Contraction Fraction; LVEDV BSA, Left Ventricular End-Diastolic Volume indexed to Body Surface Area; LVESV BSA, Left Ventricular End-Systolic Volume indexed to Body Surface Area; LVSV BSA, Left Ventricular Stroke Volume indexed to Body Surface Area; LVM BSA, Left Ventricular Mass indexed to Body Surface Area; MAF, minor allele frequency; INFO, imputation quality score; LD, linkage disequilibrium; MTAG, multitrait analysis of genome-wide association; SIFT, sorting intolerant from tolerant; PolyPhen-2, Polymorphism Phenotyping 2; CADD, combined annotation-dependent depletion; DEPICT, Data-driven Expression Prioritised Integration for Complex Traits; eQTL, expression quantitative trait loci; GTEx, genotype-tissue expression, PRS; Polygenic risk scores, MAGMA, multimarker analysis of genomic annotation; KEGG, Kyoto Encyclopedia of Genes and Genomes; IMPC, International Mouse Phenotyping Consortium; Hi-C, long-range chromatic interaction; DGIdb, drug-gene interaction database.

2. Related to point 1, there are a lot of traits and a lot of multi trait analyses. Would it make sense to limit the main results to the traits adjusted for body surface and then just talk about the traits not adjusted for body surface in the supplementary information as sensitivity analyses ? this will make it easier to keep track of the main findings. I suspect there are very few signals that would be “lost”.

We thank the reviewer for this thoughtful suggestion regarding clarity of presentation. We agree that the large number of correlated traits and multi-trait analyses can be challenging to navigate, and we carefully considered whether restricting the main text to body surface area (BSA)-adjusted traits alone would be appropriate.

However, we believe that retaining both BSA-indexed and non-indexed phenotypes in the main analyses is scientifically important because these measures could capture complementary and non-redundant aspects of ventricular structure and function. The best way to account for body size remains controversial for imaging measurements⁸. While BSA indexing remains widely used in clinical practice, several CMR studies have shown that indexing to BSA may under-recognise left ventricular hypertrophy in overweight individuals^{9,10}. Current Society for Cardiovascular Magnetic Resonance (SCMR) guidelines accept both indexed and unindexed CMR-derived parameters to enable careful interpretation in the context of body composition differences and the effects of obesity on cardiac remodelling¹¹. Consistent with this, several unindexed traits in our study identified unique genetic associations that were not detected for their BSA-adjusted counterparts (Table 1). While the total number of loci signals is small (1-5 unique loci per unindexed trait), these include novel CMR loci not found associated with any of the other 19 phenotypes – suggesting that restricting analyses to indexed traits alone would result in genuine loss of biologically meaningful signals.

To directly address the concern about redundancy and interpretability, we have already taken specific steps to prevent artificial inflation and duplicate discovery by excluding

phenotype pairs consisting of the same trait with and without BSA adjustment from the multi-trait analyses (as described in point 1 above). In addition, in response to feedback from multiple reviewers, we have expanded both the Introduction and Methods sections to provide clearer biological justification for each phenotype and to distinguish more explicitly between indexed and non-indexed measures.

Overall, our integrated framework intentionally leverages both BSA-indexed and unindexed representations to maximise biological and translational insight. Rather than moving all non-indexed traits to the Supplementary Information as sensitivity analyses, we believe that the expanded phenotype descriptions, clearer methodological justification, and improved figure labelling now provide sufficient structure for readers to interpret the main findings, while preserving the full breadth of biologically informative genetic signals.

Phenotype	Number of Unique Unindexed Loci	Number of novel CMR loci
LVEDV	3	1 (NEK4*)
LVESV	1	0
LVSV	5	2 (PRKAG1* and KLRG2)
LVM	3	1 (CTNNA3)
RVEDV	5	2 (FOXN3 , TBX2)
RVESV	6	1 (MAP3K7CL , FOXN3)
RVSV	2	0

Table 1. Unique loci counts for unindexed traits when compared against the associations of their BSA indexed equivalent phenotypes. Genes denoted with * are novel CMR loci only found associated with that unindexed phenotype out of the 20 total phenotypes.

Minor points

1. It would help readability to start sentences with written numbers e.g. Twenty one rather than 21.

We have updated the manuscript to spell out numbers when they begin sentences throughout the text.

2. Main results. There are some slightly confusing statements scattered throughout the manuscript, for example line 103 the sentence starting “Using single trait” GWAS I find confusing, although I think I can work it out, but perhaps expand to explain more clearly. In the abstract “rare variant analyses found significant burden...” is not clear .

We have clarified the wording in the Abstract and the Results sections to improve readability. In the abstract, “significant burden” has been replaced with “*statistically significant accumulation of deleterious variants*” to explicitly describe the rare variant analysis. Line 103 has been expanded to explain the workflow: “*We first performed single-trait GWAS for each phenotype. Then, to increase power, we conducted pairwise multi-trait discovery analyses by combining genetically correlated phenotypes ($r_g > 0.7$), allowing detection of additional loci associated with shared genetic variation.*”

3. Line 220 and paragraph. What is meant by “interacted” in this context ? I presume this is about the gene product encoded by the gene, not genetic variants ?

We have updated this sentence to clarify the meaning: “*Twenty-one out of the 59 genes encode proteins that interact with 76 cardiovascular drugs, among which IGF1R was prioritised at score 3.*”

4. Results rare variant analyses. Line 257 worth stating more clearly which rare variants went in, not just by frequency.

We have expanded our Results and Methods section to clarify the rare variants used in our gene burden testing. Rare variants ($MAF < 0.01$) were aggregated at the gene level using several functional masks to capture different types of predicted deleterious variation:

- Mask1: Loss-of-function (LoF) variants
- Mask2: LoF variants filtered for high confidence (LoF HC)
- Mask3: LoF plus missense variants with REVEL score > 0.5
- Mask4: LoF HC plus missense variants with REVEL score > 0.5
- Mask5: LoF plus all missense variants
- Mask6: LoF HC plus all missense variants

This strategy increases sensitivity while retaining specificity for detecting rare variant associations with ventricular phenotypes. We have updated the Methods and Results to explicitly state the use of these masks for gene-burden testing:

In the Results we have updated the sentence to:

“Rare variants ($MAF < 0.01$) from WES data were aggregated into gene units filtered by functional masks (see Methods) for gene-burden testing.”

In the Methods we have expanded to outline the functional masks:

“For the VEP annotation masks, we applied six functional masks to capture different predicted deleterious effects: loss-of-function variants (mask 1), high-confidence loss-of-function variants (mask 2), loss-of-function plus missense variants with REVEL score > 0.5 (mask 3), high-confidence loss-of-function plus missense variants with REVEL > 0.5 (mask 4), loss-of-function plus all missense variants (mask 5), and high-confidence loss-

of-function plus all missense variants (mask 6). Gene-burden testing was performed across all masks, and the global p-value (taking into account all 6 masks) was reported for each gene. Conditional analyses were performed to evaluate whether rare variant associations were independent of nearby common variants (MAF>0.01)."

5. Methods. What do the authors mean by "indexed on" ? when referring to the body surface area correction ?

By "indexed to body surface area (BSA)," we mean that the raw volumetric or mass measurements (e.g., LVEDV, LVESV, LVSV, LVM, RVEDV, RVESV, RVSV) were divided by each participant's BSA to account for differences in body size. This allows comparison of ventricular measurements across individuals of different sizes by normalising the values relative to body surface area, rather than using absolute volumes or masses. We have clarified this phrasing in the Methods section to improve readability:

"To account for inter-individual differences in body size, selected volumetric and mass phenotypes are indexed to body surface area (BSA) (LVEDV, LVESV, LVSV, LVM, RVEDV, RVESV, and RVSV), whereas ratio-based and functional measures are inherently size-normalised and therefore reported without additional BSA adjustment. Together, these 20 phenotypes were selected a priori to capture complementary and physiologically meaningful domains of ventricular size, contractile performance, myocardial remodelling, and pump efficiency across both ventricles, rather than to represent independent traits."

6. Worth spelling out minMAC first time used in methods

We have now defined minMAC in the Methods. Specifically, we now state that minMAC refers to the minimum minor allele count used in REGENIE Step 2 and briefly explain that the 20 threshold helps avoid testing extremely low-count variants; we also clarify that reported results were restricted to variants with MAF > 0.01.

We have updated this section in the Methods to:

"REGENIE step 2 analysis included setting minMAC (minimum minor allele count) of 20 to avoid testing extremely low-count variants. Following association testing, variants were filtered for downstream analyses based on minor allele frequency (MAF > 0.01) and imputation quality (INFO > 0.3), resulting in approximately 8,899,000 variants included in the final analyses. GWAS results from analysing 8,899,000 TOPmed variants (using Manhattan, QC, QQ, and regional plots) were visualised in R (v4.1.1) using packages ggplot (v3.4.2) and topR (v1.1.10)."

7. Lines 472 and the use of TOPMED section. This section of the methods is about the imputation panel ? GWAS was not “performed” in TOPMED ? check wording.

We have reworded the Methods to clarify that genotypes were imputed to the Trans-Omics for Precision Medicine (TOPMed) reference panel and that GWAS association testing was carried out on the resulting imputed genotype dosages (using REGENIE Step 2).

8. The circular manhattan plots look nice but are a little hard to read. It is not clear how the gene names map to the traits, and a colour system is challenging. Perhaps better to simply leave to the supplementary figures in straight lines to aid readability but taking up more room ? ST1 the Genetic correlations maybe better as a main figure ?

We agree that the circular Manhattan plots are visually dense and can make it difficult to follow trait-gene mappings. Following similar feedback from reviewer 2, we have replaced Figure 2 in the main manuscript with Miami-style Manhattan plots (previously shown in Supplementary Figure 2), which provide a clearer, linear representation for cross-phenotype comparison.

9. From page 26 there is a dump of data that is unreadable in the pdf version I have. Has the format been disrupted ?

The manuscript ends at page 25 so there may have been an issue with compiling the pdfs on submission. The final main figure on page 25 is the PheWAS results.

10. See point 5 but the SF manhattens are hard to read , consider placing one on each row not two and using more distinct colours.

In relation to our response to point 8, we have updated the plot to increase the readability (colours, font size, and label positioning).

11. SF3 what does “interacting associations” mean ?

This is a typo and it is now updated to “shared associations” identifying the top three phenotypes with the most shared associations to plot the three-way Venn diagram.

References

1. Liu, T. *et al.* Association Between Left Ventricular Global Function Index and Outcomes in Patients With Dilated Cardiomyopathy. *Front. Cardiovasc. Med.* **8**, 751907 (2021).

2. Reinstadler, S. J. *et al.* Prognostic value of left ventricular global function index in patients after ST-segment elevation myocardial infarction. *Eur. Heart J. - Cardiovasc. Imaging* **17**, 169–176 (2016).
3. Huang, S. *et al.* Left ventricular global function index by magnetic resonance imaging — a novel marker for differentiating cardiac amyloidosis from hypertrophic cardiomyopathy. *Sci. Rep.* **10**, 4707 (2020).
4. Arenja, N. *et al.* Myocardial contraction fraction derived from cardiovascular magnetic resonance cine images-reference values and performance in patients with heart failure and left ventricular hypertrophy. *Eur. Heart J. Cardiovasc. Imaging* **18**, 1414–1422 (2017).
5. Leibowitz, D., Zwas, D., Amir, O. & Gotsman, I. The association between myocardial contraction fraction assessed by echocardiography and mortality. *Echocardiography* **40**, 608–614 (2023).
6. Rusina, P. V. *et al.* Genetic support for FDA-approved drugs over the past decade. *Nat. Rev. Drug Discov.* **22**, 864 (2023).
7. Minikel, E. V., Painter, J. L., Dong, C. C. & Nelson, M. R. Refining the impact of genetic evidence on clinical success. *Nature* **629**, 624–629 (2024).
8. Dewey, F. E., Rosenthal, D., Murphy, D. J., Froelicher, V. F. & Ashley, E. A. Does size matter? Clinical applications of scaling cardiac size and function for body size. *Circulation* **117**, 2279–2287 (2008).
9. Riffel, J. H. *et al.* Cardiovascular magnetic resonance of cardiac morphology and function: impact of different strategies of contour drawing and indexing. *Clin. Res. Cardiol. Off. J. Ger. Card. Soc.* **108**, 411–429 (2019).
10. Brumback, L. C. *et al.* Body size adjustments for left ventricular mass by cardiovascular magnetic resonance and their impact on left ventricular hypertrophy classification. *Int. J. Cardiovasc. Imaging* **26**, 459–468 (2010).
11. Hundley, W. G. *et al.* Society for Cardiovascular Magnetic Resonance (SCMR) guidelines for reporting cardiovascular magnetic resonance examinations. *J. Cardiovasc. Magn. Reson. Off. J. Soc. Cardiovasc. Magn. Reson.* **24**, 29 (2022).

REVIEWERS' COMMENTS

Reviewer #2 (Remarks to the Author):

The authors have addressed my comments.

Reviewer #3 (Remarks to the Author):

The authors have adequately addressed my previous concerns by clarifying the rationale for the inclusion of multiple correlated CMR phenotypes, expanding the physiological definitions of imaging traits, and providing a transparent description of covariate adjustment.

One minor issue remains regarding the readability of several tables and supplementary tables, which appear to have formatting issues following conversion from Excel to PDF. These should be revised.

We thank the reviewer for all their feedback throughout this process. To conform also to the editorial requirements, all tables are now supplementary data files, which we believe should resolve any conversion issues.

Reviewer #4 (Remarks to the Author):

Many thanks to the authors for addressing the questions. There is still a formatting issue in the main table in my version but I will leave that with the editors and authors.

We appreciate the reviewer's thoughtful feedback and guidance throughout this process. We have since updated table 1 to be a supplementary data table, which should resolve any formatting issues.